

# Southern California Megacity CO$_2$, CH$_4$, and CO flux estimates using remote sensing and a Lagrangian model

Jacob K. Hedelius[1,2], Junjie Liu[3,1], Tomohiro Oda[4,5], Shamil Maksyutov[6], Coleen M. Roehl[1], Laura T. Iraci[7], James R. Podolske[7], Patrick W. Hillyard[8], Debra Wunch[2], and Paul O. Wennberg[1,9]

[1]California Institute of Technology, Division of Geology and Planetary Science, Pasadena, California, USA
[2]University of Toronto, Department of Physics, Toronto, Ontario, Canada
[3]Jet Propulsion Laboratory, California Institute of Technology, Pasadena, California, USA
[4]Global Modeling and Assimilation Office, NASA Goddard Space Flight Center, Greenbelt, MD, USA
[5]Goddard Earth Sciences Technology and Research, Universities Space Research Association, Columbia, MD, USA
[6]Center for Global Environmental Research, National Institute for Environmental Studies, Tsukuba, Ibaraki, Japan
[7]NASA Ames Research Center, Mountain View, CA, USA
[8]Bay Area Environmental Research Institute, Petaluma, CA
[9]Division of Engineering and Applied Science, California Institute of Technology, Pasadena, CA

**Correspondence:** Hedelius, J. K. (jacob.hedelius@utoronto.ca)

**Abstract.** We estimate the overall CO$_2$, CH$_4$, and CO flux from the South Coast Air Basin using an inversion that couples Total Carbon Column Observing Network (TCCON) and Orbiting Carbon Observatory-2 (OCO-2) observations, with the Hybrid Single Particle Lagrangian Integrated Trajectory (HYSPLIT) model, and the Open-source Data Inventory for Anthropogenic CO$_2$ (ODIAC). Using TCCON data we estimate the direct net CO$_2$ flux from the SoCAB to be $139 \pm 35$ Tg CO$_2$ yr$^{-1}$ for the

study period of July 2013–August 2016. We obtain a slightly lower estimate of $118 \pm 29$ Tg CO$_2$ yr$^{-1}$ using OCO-2 data. These CO$_2$ emission estimates are in general agreement with previous work. Our net CH$_4$ ($325 \pm 81$ Gg CH$_4$ yr$^{-1}$) flux estimate is slightly lower than central values from previous top-down studies going back to 2010 (342–440 Gg CH$_4$ yr$^{-1}$). CO emissions are estimated at $555 \pm 136$ Gg CO yr$^{-1}$, much lower than previous top-down estimates (1440 Gg CO yr$^{-1}$). Given the decreasing emissions of CO, this finding is not unexpected. We perform sensitivity tests to estimate how much errors in the prior,

errors in the covariance, different inversions schemes or a coarser dynamical model influence the emission estimates. Overall, the uncertainty is estimated to be 25 %, with the largest contribution from the dynamical model. The methods described are scalable and can be used to estimate direct net CO$_2$ fluxes from other urban regions.

## 1 Introduction

About 43 % of global anthropogenic carbon dioxide (CO$_2$) emissions come directly from urban areas, and urban final energy

use accounts for about 76 % of CO$_2$ emissions (Seto and Dhakal, 2014). Associations of cities that recognize their significant emissions of CO$_2$ to the atmosphere—such as the C40 Cities Climate Leadership Group (C40)—seek to reduce their greenhouse gas (GHG) emissions and develop local resilience to changing climate. There is a need to track long-term anthropogenic GHG emissions from urban areas to aid urban planners and ensure commitments are met.





Tracking emissions from a top-down (TD) perspective requires observations. Various networks, such as the Total Carbon Column Observing Network (TCCON), and the National Oceanic and Atmospheric Administration (NOAA) Earth System Research Laboratory (ESRL) in situ $CO_2$ network can aid in long-term measurements, but are too sparse to track emissions from 100+ cities. Some urban areas have ground-based networks (e.g., Lauvaux et al., 2016; Shusterman et al., 2016; Verhulst

et al., 2017; Mitchell et al., 2018). Significant progress has been made in minimizing the cost, deployment time, and data delivery from these networks. However, they still require a significant number of personnel hours and are difficult to scale-up to many (100+) areas for long-term observations. Urban observation networks can provide finer spatial and temporal details on emission sources, but space-based observations are likely the only way to track emissions from a large number of cities.

Within the past 10 years, 2 satellites have been shown to have high precision (better than 1 ppm) small footprint ($< 100 \, \mathrm{km}^2$)

$CO_2$ observing capabilities, including the Greenhouse Gases Observing Satellite (GOSAT, in orbit 2009) and the Orbiting Carbon Observatory-2 (OCO-2, in orbit 2014). Several other satellites are planned or are already in orbit with this same potential. Combined, OCO-2 and GOSAT can cover about 1 % of the Earth's surface every 3 days, and though this is only a small fraction, it is unprecedented. Other missions such as TanSat (in orbit, 2016), GAS onboard FY-3D (in orbit, 2017), GOSAT-2 (expected, 2018), and GeoCARB (expected, 2022) may further bolster coverage. Space-based observations of methane ($CH_4$)

have been made from GOSAT and the TROPOspheric Monitoring Instrument (TROPOMI, in orbit 2017), and will be made from the planned GOSAT-2 and GeoCARB missions. Carbon monoxide (CO) is measured using Measurements of Pollutants in the Troposphere (MOPITT, in orbit 1999), TROPOMI, and will be from GOSAT-2. There is a need to assimilate these data in inversion schemes to determine urban fluxes, and long-term trends. Ideally, such a scheme will be efficient enough to scale up and to incorporate future datasets.

We test trajectory-based inversion schemes to see if they can reproduce known emissions (from inventories and previous studies) from the California South Coast Air Basin (SoCAB). Our goal is not to apportion spatially, but rather to come up with a single number for the total flux, and an estimate of uncertainty. Fluxes from this urban area (pop. $\sim$16.3 million) have been studied extensively, and it provides a test bed to evaluate methods. We discuss the components used to build our inversion in Sect. 2. Typical urban enhancements are described in Sect. 3. Fluxes of $CO_2$, CO, and $CH_4$ using TCCON data, and of

$CO_2$ using OCO-2 data are discussed in Sect. 4 along with sources of uncertainty. In Sect. 5 we discuss emission ratios, which can also be used to validate our results. We conclude by summarizing uncertainty, mentioning expansions, and areas of improvement in Sect. 6.

## 2   Data sources and methods

### 2.1   Column-averaged dry-air mole fraction observations

We use column-averaged dry-air mole fraction observations (denoted $X_{gas}$) to tie model abundances to fluxes. Column-averages are calculated by dividing the retrieved amount of the gas of interest (in molecules $\mathrm{cm}^{-2}$) by the retrieved total column of dry-air (in molecules $\mathrm{cm}^{-2}$). $X_{gas}$ values are less sensitive to changes in surface pressure and water vapor than total column amounts in units of molecules $\mathrm{cm}^{-2}$ (Wunch et al., 2015).



Data are obtained from the TCCON and OCO-2. We use TCCON data from the California Institute of Technology (Caltech) site in Pasadena, California (Wennberg et al., 2014), as well as the NASA Armstrong Flight Research Center (AFRC) site near Lancaster, California (Iraci et al., 2014). Values of $X_{CO_2}$, $X_{CO}$, and $X_{CH_4}$ were generated using the operational GGG2014 algorithm (Wunch et al., 2015). The Caltech site (34.136° N, 118.127° W, 240 m a.s.l.) is located in an urban environment

within the SoCAB. As the name implies, the SoCAB is a basin surrounded by mountains, except towards the southwest which boarders the Pacific Ocean. AFRC (34.960° N, 117.881° W, 700 m a.s.l.) is located outside the basin ∼100 km to the north in a much more sparsely populated area. Because of the lower population density, the AFRC is often considered a 'background' site. However, depending on airflow patterns recent emissions from the SoCAB may be observed at the AFRC so we use the term 'background' loosely to indicate where lower concentrations are typically observed. Coincident data from both sites are

available from July 2013–August 2016 after which the AFRC instrument was relocated. In total, there are 5,355 paired hourly averaged observations on 783 days.

OCO-2 data are available starting September 2014 when the instrument began its nominal operational mission (OCO-2 Science Team et al., 2017). Here, we use $X_{CO_2}$ data generated using the NASA Atmospheric $CO_2$ Observations from Space (ACOS) version 8r algorithm (Crisp et al., 2012; O'Dell et al., 2012). We also do a partial analysis on V7r data for comparison

with past studies that used these data with a focus on the SoCAB (Hedelius et al., 2017a; Schwandner et al., 2017). Because OCO-2 is in a sun-synchronous orbit with an equatorial crossing time of around 1 pm local solar time, all observations of the SoCAB are made in the early afternoon. OCO-2 has 8 longitudinal pixels, with a footprint of ∼3 km$^2$ each. To reduce over-weighting target mode observations, OCO-2 data are gridded to $0.01° \times 0.01°$. There are 6,098 pre-averaged OCO-2 observations on 29 different overpass days when the AFRC TCCON site also collected background observations before filtering.

In Appendix A we describe filtering, background subtraction, boundary conditions, and our accounting for averaging kernels. In short, we determine enhancements of various gases ($\Delta X_{gas}$) by finding the difference between observations within the basin (either the Caltech TCCON, or OCO-2) compared with the AFRC TCCON site.

## 2.2 A priori flux estimates

Our flux estimate involves scaling the a priori spatial inventory, or sub-regions of the prior up or down to reduce the measured−model

mismatch. More important than the total prior absolute flux is the distribution of sources. Hestia-LA v2.0 is likely the most accurate spatiotemporal inventory for the SoCAB, however it is not available globally. EDGAR (Emissions database for global atmospheric research, EC-JRC/PBL (2009)) and FFDAS v2.0 (Fossil Fuel Data Assimilation System, Asefi-Najafabady et al. (2014)) are available globally at a 0.1° resolution. We use the year 2016 version of the Open-source Data Inventory for Anthropogenic $CO_2$ (ODIAC2016) which is available globally at a resolution of 30 arcseconds from 2000–2015 (Oda and Maksyutov,

2011, 2015; Oda et al., 2018). We also compare total SoCAB emissions from the 2015 version of ODIAC (ODIAC2015) which is based on a projection of the Carbon Dioxide Information Analysis Center (CDIAC) country total emissions. For the Indianapolis region, Lauvaux et al. (2016) noted little difference in the aggregate inversion flux when using ODIAC compared with Hestia. We assume that 2015 emissions are identical to those in 2016. A generic temporal hourly scaling factor product (TIMES - Temporal Improvements for Modeling Emissions by Scaling) available at a $0.25° \times 0.25°$ can be applied to spa-



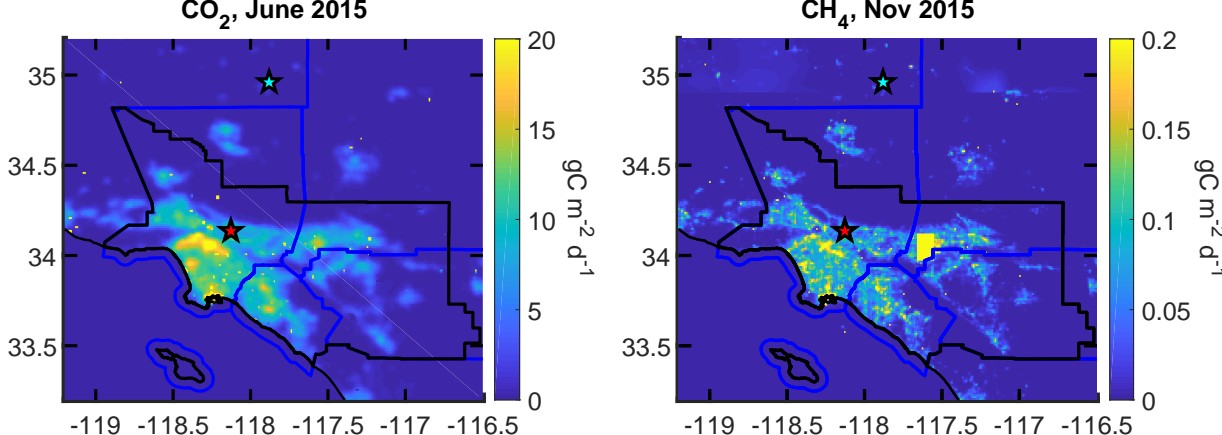

**Figure 1.** A priori flux maps for $CO_2$ (left) and $CH_4$ (right) for select months. The same spatiotemporal prior for $CO_2$ (ODIAC2016) was used for CO, but scaled to 1 % on a per mole basis. The ODIAC product was downscaled to $0.01°$ resolution. The methane prior was created based on point sources, total emissions, and the population distribution.

tial inventories such as ODIAC to improve temporal emissions (Nassar et al., 2013). However, TIMES has a single peak for mid-day emissions, which is inconsistent with morning and afternoon rush hour periods in the SoCAB. We instead use the Hestia-LA v1.0 weekly profile reported by Hedelius et al., (2017a, Fig. 2 therein) which has both morning and afternoon rush hour peaks. We downscale the ODIAC to a $0.01° \times 0.01°$ grid over the domain 121.5°W–114.5°W and 30.5°N–37.5°N. This

same prior is used for CO, but total emissions are 1 % of $CO_2$ emissions on a molar basis (0.6 % of mass). Figure 1 shows the ODIAC2016 prior for one month.

We make our own $0.01° \times 0.01°$ methane prior using landfills, nightlights, expected total emissions, and the Harvard-Environmental Protection Agency (EPA) United States (U.S.) inventory (Maasakkers et al., 2016) shown in Fig. 1. A more detailed $CH_4$ inventory is also available for the SoCAB, which we do not use because it would be difficult to scale globally

(Carranza et al., 2018). First, we distribute emissions from landfills as point sources (available 2010–2015, https://ghgdata.epa. gov/ghgp/main.do) and use 2015 emissions for 2016. Emissions from the Puente Hills landfill were doubled because the EPA estimate (average $13.6\,\mathrm{Gg\,CH_4\,yr^{-1}}$) is low compared to previous estimates of $34\,\mathrm{Gg\,CH_4\,yr^{-1}}$ (Peischl et al., 2013). After doubling Puente Hills emissions, EPA total SoCAB ($144\,\mathrm{Gg\,CH_4\,yr^{-1}}$) and Olinda Alpha ($13.5\,\mathrm{Gg\,CH_4\,yr^{-1}}$) landfill emissions are similar enough to other studies ($164\,\mathrm{Gg\,CH_4\,yr^{-1}}$ and $12.5\,\mathrm{Gg\,CH_4\,yr^{-1}}$ respectively, Peischl et al., 2013) that we do

not double emissions from other landfills in the SoCAB. Chino dairy emissions were added in as a $\sim 0.1° \times 0.1°$ source (Chen et al., 2016; Viatte et al., 2017). Outside of the SoCAB $CH_4$ manure and enteric fermentation were added from the $0.1° \times 0.1°$ Harvard-EPA inventory (Maasakkers et al., 2016). SoCAB emissions are assumed to sum to $400\,\mathrm{Gg\,CH_4\,yr-1}$ based on the



work of Wunch et al. (2016), and the rest of the emissions were distributed based on population which was assumed to correspond with the January 2017 Suomi NPP nightlights (15 arcseconds). An average monthly trend was included based on results of Wong et al. (2016), and emissions were assumed to be constant on a monthly timescale. Because the Aliso Canyon leak effectively doubled the SoCAB $CH_4$ emissions for its duration from 23 October 2015 to 11 February 2016 (Conley et al.,

2016), it was also added as a point source.

We use various publicly available statistics to get a sense of annual $CO_2$ emissions from the SoCAB. Literature estimates range from $99\,\mathrm{Tg}\,CO_2\,\mathrm{yr}^{-1}$ (Vulcan, Fischer et al., 2017) to $211\,\mathrm{Tg}\,CO_2\,\mathrm{yr}^{-1}$ (EDGAR v4.0, as reported by Wunch et al., 2009). Table 1 lists statistics for the SoCAB. We assume the non-residential natural gas (NG) use is for industry or power accounted for in the EPA inventory. Because most of the food consumed in the SoCAB is grown outside the basin, such as in

the Midwestern U.S. and Central Valley (CV), there is a $CO_2$ return flux to the croplands from both human respiration and food waste. In the U.S. 60 million metric tonnes (MMT) of food are lost annauly at the retail and consumer levels compared with 129 MMT consumed (Dou et al., 2016), roughly one-third of all food calories (not counting inedible food related biomass). Presumably, most food waste decomposition would be accounted for in EPA landfill emissions. However, $CO_2$ emissions from food waste could be underestimated if food waste is composted, if there were unaccounted for methanotrophs, or if aerobic

respiration is significantly underestimated (e.g., from rapid decomposition while still exposed to oxygen) which would decrease the $CH_4:CO_2$ emission ratio commonly assumed to be unity for managed landfills on a per mole basis (RTI, 2010). Thus, we add $30\,\%$ to human respiration emissions of $917\,\mathrm{g}\,CO_2\,\mathrm{d}^{-1}\,\mathrm{person}^{-1}$ (Prairie and Duarte, 2007) for food waste losses. We assume the flux from vegetation is balanced (i.e., no net change in plant biomass or soil carbon) within the basin. Based on these various statistics we estimate a bottom-up net flux on the order of $110\,\mathrm{Tg}\,CO_2\,\mathrm{yr}^{-1}$ from the SoCAB.

## 2.3 Dynamical models

A dynamical model is needed in conjunction with the a priori flux estimates to generate forward model $X_{\mathrm{gas}}$ enhancements. This may be as complex as a custom high-resolution Weather Research and Forecasting (WRF) model (e.g., Lauvaux et al., 2016) or as simple as an average mixed layer wind velocity (e.g., Chen et al., 2016). Our model uses Lagrangian trajectories driven by existing, archived forecast or reanalysis datasets.

An advantage of archived model data is there is no need to run an Eulerian model first, and they are more accessible to a broader community. However, taking existing results without model evaluation may propagate hidden errors and biases which could influence flux results. Archived data usually have coarser spatiotemporal resolutions than custom models, and cover larger domains than the area of interest. Custom runs allow models to be parameterized differently and nudged to reduce the measured−model mismatch for the regions of interest.

We use the North American Mesoscale Forecast System (NAM) at $12\,\mathrm{km}$ resolution ($3\,\mathrm{hr}$ temporal) from the NOAA data archive as the primary model source. NAM is run with a non-hydrostatic version of the WRF at its core with a Mellor–Yamada–Janjić planetary boundary layer (PBL) scheme (Coniglio et al., 2013). Estimates of model error are described in Appendix B. Though NAM data are only available over North America, other archived models are available at lower resolution with global coverage (e.g., the Global Data Assimilation System (GDAS) $0.5\,^{\circ}$, $3\,\mathrm{hr}$ product). The NOAA ESRL recently



**Table 1.** Statistics for the SoCAB

| Description | Value | Description | Value |
|---|---|---|---|
| Population | 16.3 million | Motor gasoline[d,e] | 6.8 B gal yr$^{-1}$ |
| Population (of CA) | 42 % | | 60 Tg CO$_2$ yr$^{-1}$ |
| Area | 17,100 km$^2$ | Diesel fuel[d,e] | 1.3 B gal yr$^{-1}$ |
| Direct U.S. GHG | 2 % | | 13 Tg CO$_2$ yr$^{-1}$ |
| Direct global GHG | 0.25 % | Human respiration + food waste[f] | 8 Tg CO$_2$ yr$^{-1}$ |
| Cities[a] | 162 | Natural gas total (residential)[g,h] | 430 (190) TBTU |
| Vehicle miles (VM)[b] | 140 B yr$^{-1}$ | | 23 (10) Tg CO$_2$ yr$^{-1}$ |
| Passenger VM emissions[c,d] | 55 Tg CO$_2$ yr$^{-1}$ | EPA industry/power/waste[i] | 20.5 Tg CO$_2$ yr$^{-1}$ |
| Truck VM emissions[c,d] | 12 Tg CO$_2$ yr$^{-1}$ | Air traffic est.[i] | 0.5 Tg CO$_2$ yr$^{-1}$ |
| | | Cargo ships est.[i] | 2 Tg CO$_2$ yr$^{-1}$ |

Most of these values are approximations. [a]http://www.aqmd.gov/home/about/jurisdiction
[b]http://www.dot.ca.gov/hq/tsip/hpms/datalibrary.php [c]Assuming 95 % of miles light duty vehicles with 21.5 mile per gallon (MPG) fuel
efficiency, and 5 % trucks with 5.8 MPG (https://www.fhwa.dot.gov/policyinformation/statistics/2013/, VM-1) [d]Vehicle miles and fuel
emissions are independent estimates. [e]http://www.cdtfa.ca.gov/taxes-and-fees/spftrpts.htm [f]Based on emissions of
$1.3 \times 917$ g CO$_2$ d$^{-1}$ person$^{-1}$ (Prairie and Duarte, 2007) [g]http://www.ecdms.energy.ca.gov/gasbycounty.aspx
[h]https://www.epa.gov/sites/production/files/2015-07/documents/emission-factors_2014.pdf [i]Emissions within or near geographical SoCAB
boundaries only

began publicly releasing 3 km, 1 hr archived data from the High Resolution Rapid Refresh (HRRR) model that covers the U.S. (Benjamin et al., 2016). This product holds the potential to improve flux estimates at smaller scales.

We use HYSPLIT-4 (Hybrid Single Particle Lagrangian Integrated Trajectory-4; Stein et al., 2015) with the 3 archived NOAA data products described above. Our base method is to use mean 48 hr back trajectories with NAM 12 km for the

5   lowest 20 % of the atmosphere, which we assume is the only part of the atmosphere enhanced with local emissions at the measurement site. Trajectories are equally spaced in pressure every 0.3 % of the column. By comparison, the GDAS model takes $0.71 \pm 0.18\,(1\sigma)$ times as long to run, and the HRRR model takes $33.3 \pm 7.1\,(1\sigma)$ times as long. Because HRRR takes substantially longer, we only run it for a subset of months—July, October 2015, and January, April 2016. Other studies (e.g., Janardanan et al., 2016; Fischer et al., 2017) used multiple particles released at each level. We assume that over the multi-

10  year time series the ensemble of mean trajectories is, on average, representative of the upwind influences on the receptor sites without the additional turbulence term.

Figure 2 shows back trajectories for one layer and 2 different times that end at the observation sites. Trajectories from multiple vertical levels are combined to determine residence times or footprints as described in Appendix C. HYSPLIT shows 3 major origins for air at the Caltech site. The primary source is from over the ocean and over downtown Los Angeles (southwest).





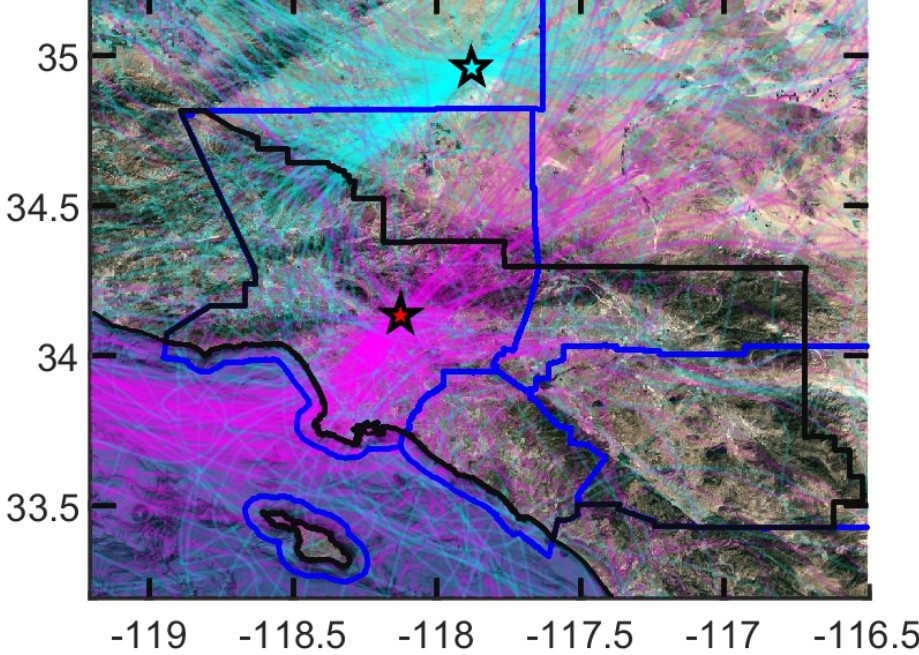

**Figure 2.** HYSPLIT 400 m a.g.l. back trajectories for NAM 12 km for 2015. For each day trajectories are shown ending at the 2 different TCCON receptor sites at 14:00 (UTC-7). Magenta trajectories end at Caltech. Cyan trajectories end at AFRC.

The second major source is from the Mojave desert (northeast), and the third source is from the Central Valley (northwest, see Fig. E1).

## 2.4 Inverse methods for comparing measured to model data

Different schemes can be applied to reduce the measured−model mismatch. One of the simplest is to find the ratio between the average enhancements in the observations compared with the forward model and then to scale the prior based on this ratio. Bayesian inversions are more complex, but can also improve information on the spatial distribution and intensity of fluxes (e.g., Turner et al., 2016; Lauvaux et al., 2016); they can be solved by analytical or adjoint methods (Rodgers, 2000; Kopacz et al., 2009). Different cost functions can be used, which might change the results. Here we test 3 different methods. The first is a Kalman filter (described in Appendix D) which is computationally cheap, but has only one degree of freedom. For scaling retrievals, using too few degrees of freedom can cause the results to be heavily weighted by the largest model results relative to the observations (Appendix D2). We also use Bayesian inversions based on the methods of Rodgers (2000) (described in Appendix E). One Bayesian inversion is based on a non-linear forward model with 40 different scaling factors (Eq. E2), and the other is a linear forward model with up to nearly 35,000 scaling factors (Eq. E3), though only a fraction (< 1000) of these





are used. Because of potential bias in the first two methods, we focus on the linear forward model. Uncertainty estimates are stated for the linear forward model while disregarding the other methods.

## 2.5 Summary: Data sources and methods

In summary, we have 4 sets of observations of $X_{gas}$ differences: Caltech TCCON − AFRC TCCON ($CO_2$, $CH_4$, and CO), and OCO-2 − AFRC TCCON ($CO_2$). We use one gridded spatiotemporal inventory for $CO_2$ and CO (ODIAC2016, with a weekly pattern for hourly emissions), and one gridded spatiotemporal inventory for $CH_4$ (Sect. 2.2). HYSPLIT is run with three dynamical models for the Caltech TCCON − AFRC TCCON differences (GDAS $0.5°$, NAM 12 km, and HRRR 3km for a subset), and is run with NAM 12 km for the OCO-2 − AFRC TCCON differences. Three different inversion techniques are used including a Bayesian inversion with a linear forward model, a Bayesian inversion with a non-linear forward model, and a Kalman filter. Unless specified, values reported are from the Caltech TCCON − AFRC TCCON difference with the NAM 12 km model and the Bayesian inversion with the linear forward model.

## 3 Typical $X_{gas}$ enhancements

Several previous studies have discussed the SoCAB $X_{CO_2}$, $X_{CH_4}$, and $X_{CO}$ enhancements from local anthropogenic activity (Wunch et al., 2009; Kort et al., 2012; Janardanan et al., 2016; Wunch et al., 2016; Hedelius et al., 2017a; Schwandner et al., 2017). There have also been several studies which have discussed enhancements noted from the CLARS (California Laboratory for Atmospheric Remote Sensing). CLARS has a viewing geometry that is more sensitive to the mixing layer than TCCON and nadir-viewing satellites, which leads to larger typical enhancements in $CO_2$ and $CH_4$ (Wong et al., 2015, 2016). For comparability we exclude enhancements from CLARS and in situ observations (e.g., Verhulst et al., 2017) in this section. Kort et al. (2012) noted that observing changes in typical $X_{gas}$ enhancements from space-borne instruments can provide a first order estimate of how local emissions have changed year-to-year. This requires similar year-to-year ventilation patterns, and sufficiently large and representative sample sizes which is becoming less of an issue as more space-based observations become available. Changes in $X_{gas}$ enhancements can provide a first-order estimate of how much local emissions have decreased without the need for a full inversion.

Table 2 lists $X_{CO_2}$ enhancements observed over the SoCAB compared to an external background. An instrument with a smaller footprint (e.g., OCO-2, about 1.3 km×2.25 km) could observe a wider range of $X_{CO_2}$ enhancements than an instrument with a larger footprint (e.g., GOSAT, about 10.5 km diameter). However, the footprint size should not affect the average enhancement over a domain much larger than an individual footprint. In Fig. 3 are histograms of enhancements for all dates of this study. Most enhancements are on order of 2–3 ppm except for those from the recently published paper by Schwandner et al. (2017), which are about double. Though their enhancements are within the range of $\Delta X_{CO_2}$ enhancements in the v7r and v8r histograms in Fig. 3 (bottom row), they are atypical. Their results are likely atypically large because of dynamics on the two particular dates analyzed, and do not include enough data to determine typical enhancements, trends, and source and sink attribution. We disagree with their conclusions that these values are in agreement with Kort et al. (2012) and that TCCON



**Table 2.** SoCAB $X_{CO_2}$ enhancements.

| Citation | Observations | $\Delta X_{CO_2}$ (ppm) |
|---|---|---|
| Kort et al. (2012) | GOSAT-ACOS v2.9 | $3.2 \pm (1.5)\,(1\,\sigma)$ |
| Janardanan et al. (2016) | GOSAT | $2.75 \pm (2.86)\,(1\,\sigma)$ |
| Hakkarainen et al. (2016) | OCO-2 v7r | $\sim$2–2.5[a] |
| Hedelius et al. (2017a) | OCO-2 v7r & TCCON | $2.4 \pm (1.5)\,(1\,\sigma)$ |
| | TCCON, v2014 | $2.3 \pm (1.2)\,(1\,\sigma)$ |
| Schwandner et al. (2017) | OCO-2 v7r | 4.4–6.1 |
| This study[b] | OCO-2 v8r & TCCON | $2.1 \pm (1.7)\,(1\,\sigma)$ |
| | TCCON, v2014 | $2.7 \pm (1.4)\,(1\,\sigma)$ |

[a]Qualitative estimate based on Fig. 1 and Supplemental Fig. 3 therein. [b]We modified the boundary condition compared to our previous work (see Appendix A), values are for 14:00 (UTC-7).

validates this high of a typical SoCAB enhancement. Their conclusion that seasonal variations are 1.5–2 ppm does appear to be supported by previous work (Hedelius et al., 2017a). However, their full attribution of the seasonal cycle to biospheric processes within the basin is not supported by the findings of Newman et al. (2016) who found the excess $CO_2$ from the biosphere only varied from 8 % (summer) to 16 % (winter) of fossil fuel excess. More likely the changing enhancement reflects a small

change in the biosphere, and most importantly, seasonal differences in the basin ventilation.

Models that assimilate only global in situ (i.e., no total column) $CO_2$ data are biased by only about $\pm 1$ ppm ($1\sigma \sim 1$ ppm) compared with TCCON observations (Kulawik et al., 2016). This highlights the need to understand bias and uncertainty in total column observations to the order of a few tenths of a ppm or better to provide new information. The TCCON-predicted bias uncertainty is 0.4 ppm or less (<0.1 %). A long-term $CO_2$ reduction goal is to reach 20 % of 1990 levels by 2050. This is

10 about a 2–3 % decrease per year assuming a constant reduction. Thus a 0.4 ppm bias is on order of 4–9 years worth of emission reductions.

## 4 SoCAB flux estimates

### 4.1 Carbon dioxide

Our flux estimate of $CO_2$ using the TCCON sites and linear model (Eq. E3) is $139 \pm 35$ TgCO$_2$ yr$^{-1}$. An error assessment is

15 described in Sec. 4.3. This estimate is shown, along with estimates from past studies, in Fig. 4. Our estimate is comparable to Vulcan (Brioude et al., 2013; Fischer et al., 2017), Hestia-LA v1.0, EDGAR v4.2, and the California Air Resources Board (CARB) 2017 population scaling estimates, within uncertainty. Our result is also in excellent, but perhaps fortuitous agreement with Ye et al. (2017) who estimated emissions by comparing OCO-2 observations with forward model results from a WRF-Chem model. However, our result differs significantly from previous top-down (TD) estimates from aircraft flights, EDGAR





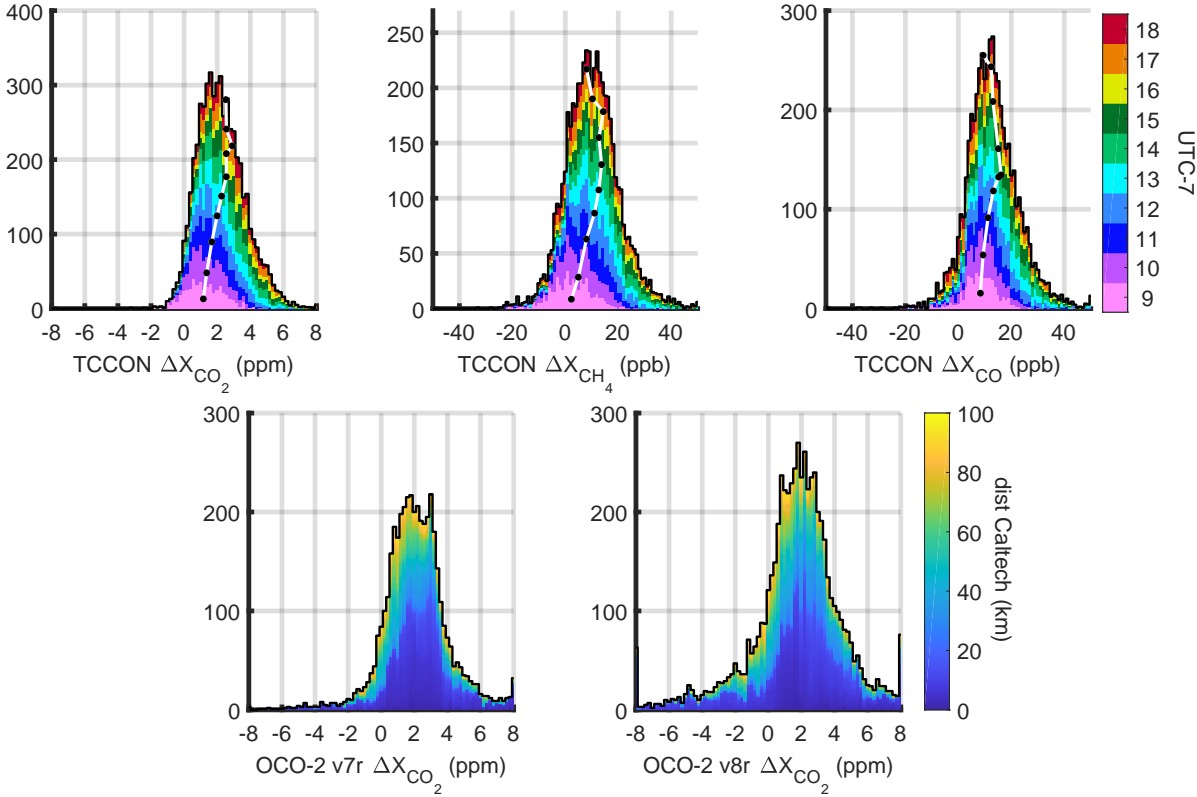

**Figure 3.** Histograms of $X_{gas}$ enhancements observed in the SoCAB. Data are averaged for $\pm 30$ minutes centered on the hour. Top row: Enhancements are defined as Caltech TCCON observations minus AFRC TCCON observations (Appendix A). Colors represent the hour of day, and white lines with black dots in the top row are hourly medians. Enhancements peak in early afternoon from morning rush hour emissions getting transported from downtown Los Angeles (southwest) to Caltech, and from mixed layer dynamics. Bottom row: Enhancements are OCO-2 observations minus AFRC TCCON observations. Colors represent the distance from the Caltech TCCON site. Histograms are for all dates of this study (Section 2.1).

v4.0 (as reported by Wunch et al., 2009), and CARB 2011. Between 2011 and 2012 CARB changed how bunker fuels and aircraft emissions were reported for the state, which caused a significant decrease in reported emissions. Our posterior estimate is larger than ODIAC2016, which is slightly less than ODIAC2015. The ODIAC2016 is based on disaggregation of CDIAC national total emissions. Thus, unlike locally developed emission inventories the interannual variations in subnational emissions are driven by the national emission trends. ODIAC could be low from incorrectly distributing to rural areas. Blooming effects (exaggerating the extent of cites due to coarse gridded spatial resolution and indirect or non-electrical lights) in the underlying nightlight data fields in ODAIC could contribute to an incorrect distribution.



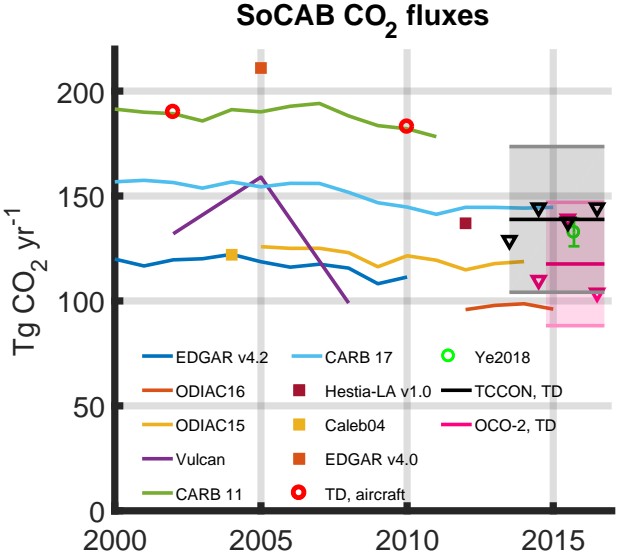

**Figure 4.** Estimates of SoCAB $CO_2$ fluxes (annual estimates from TCCON are shown as black triangles and OCO-2 are shown in pink) compared with previous studies. Top-down (TD) flight estimates are from Brioude et al. (2013). TD estimate from Ye et al. (2017) is based on OCO-2 observations and 5 % random uncertainty has been added. The Hestia-LA v1.0 estimate was inferred after a forward implementation into a WRF model (Hedelius et al., 2017a). EDGAR v4.2, ODIAC, and CARB emissions were calculated from databases. EDGAR v4.0 value was reported by Wunch et al. (2009). All other values were found in a literature review.

Most of the estimates from previous studies include only emissions from fossil fuel use. We have not separately accounted for biospheric uptake (emissions) in the model, and if it is significant, the anthropogenic flux would be larger (smaller) than our net estimate. In the GEOS-Chem model described by Liu et al. (2017) the nearby ocean is a neutral to weak sink, likely from biological activity.

5    OCO-2 provides better spatial coverage than TCCON (Fig. 5), and the orbit tracks can change longitudinally with season or when the spacecraft moves for collision avoidance. However, observations only occur at the same local solar time, and are days to weeks apart. The estimate using OCO-2 data is slightly lower at $118\pm29$ Tg$CO_2$ yr$^{-1}$. This value varies by up to 43 Tg$CO_2$ yr$^{-1}$ depending on filtering methods (e.g., warn levels, WL). Warn levels are a global metric of data quality, where WLs less than or equal to (0, 1, 2, 3, 4, 5) correspond to about (50 %, 60 %, 70 %, 80 %, 90 %, 100 %) of data passing in V8r,
10    and larger WLs generally correspond to less reliable data. After excluding just the 2 largest WLs, the total net flux varies by only 23 Tg$CO_2$ yr$^{-1}$ when applying additional filters.

## 4.2 CH$_4$ and CO

Using the same methodology we estimate a $CH_4$ flux of $325\pm81$ Gg$CH_4$ yr$^{-1}$. This is less than the estimate by Wunch et al. (2009), but similar to estimates from CARB (Fig. 6). CARB-based $CH_4$ fluxes for just the SoCAB were estimated





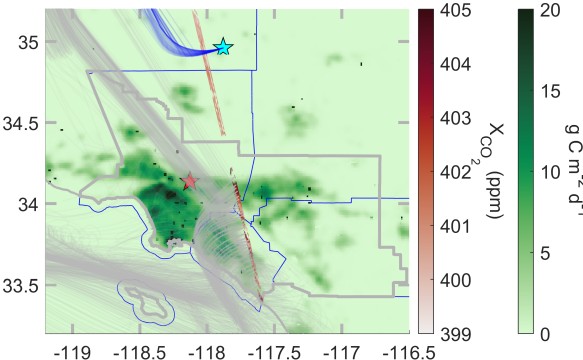

**Figure 5.** A visualization of OCO-2 observations and the forward model used in the flux inversion on June 20, 2015. The nadir track is shown starting at the bottom and $\sim 117.6°\,$W and going towards the northwest. Observations are overlaid on the ODIAC prior at 14:00 (UTC-7). For every 5th sounding the set of backtrajectories is shown.

by subtracting Agriculture and Forest emissions (53–61 % of total depending on version and year), and out-of-state electricity generation (0–0.1 %). The remaining flux was scaled by 42 % based on the population of the SoCAB, and 5 % of the Agriculture and Forestry emissions were added back in. Our estimate is lower than previous estimates of $CH_4$ fluxes using in situ (tower and aircraft) data (Hsu et al., 2010; Wennberg et al., 2012; Peischl et al., 2013; Wecht et al., 2014; Cui et al., 2015).

We also estimate a CO flux of $555\pm136\,\mathrm{Gg\,CO\,yr^{-1}}$. This is significantly less than the estimates by Wunch et al. (2009) of $1,400\pm300\,\mathrm{Gg\,CO\,yr^{-1}}$ from Aug 2007–June 2008, and the estimate of $1,440\pm110\,\mathrm{Gg\,CO\,yr^{-1}}$ by Brioude et al. (2013) for summer 2010. Wunch et al. (2009) used a tracer-tracer relationship where the assumed $CO_2$ was likely too large ($191\,\mathrm{Tg\,CO_2\,yr^{-1}}$). When their results are scaled down based on our posterior $CO_2$ fluxes ($118$–$139\,\mathrm{Tg\,CO_2\,yr^{-1}}$), the CO flux is $830$–$970\,\mathrm{Gg\,CO\,yr^{-1}}$ which is in better agreement with the CARB inventory. The CARB CO inventories, specific to the SoCAB have decreasing CO

emissions; part of the difference could be from different observation periods. CARB2017 emissions are $581\,\mathrm{Gg\,CO\,yr^{-1}}$ for 2015.

### 4.3 Sensitivity tests and error assessment

For a single estimate of the SoCAB flux, we have a sufficiently large sample that random uncertainty is small. This is supported by a bootstrap analysis where we select a random subset of data equal in size to the original $n = 200$ times (Efron and Gong,

1983). The random uncertainty estimate is $8\,\mathrm{Tg\,CO_2\,yr^{-1}}$ ($2\sigma$), or about 6 %. Persistent biases from a priori flux uncertainty, model errors, observation biases including boundary conditions, and poorly chosen initial values are more detrimental to our flux estimate.

Several variables ($\mathbf{x}_a$, $\mathbf{S}_\epsilon$, $\mathbf{S}_a$) need initial values (see Appendix E3), and how these are chosen can affect the final flux calculated. We evaluate 4 sensitivity tests (Fig. 7). For the first test, we filter out data where the observations differ from the





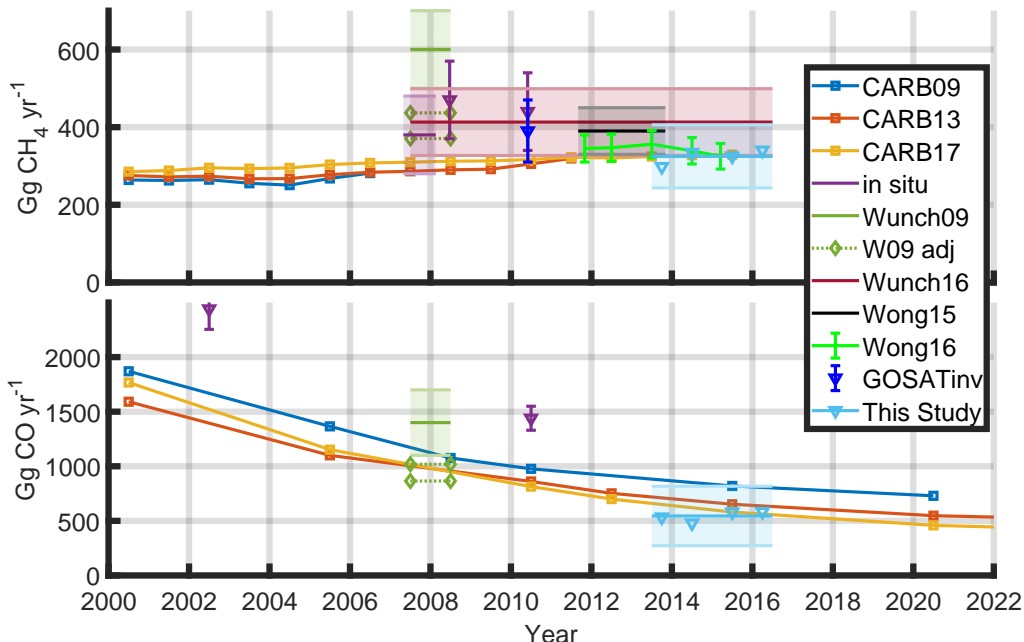

**Figure 6.** SoCAB $CH_4$ and CO flux estimates. Annual estimates from this study are shown as light blue triangles. Brioude et al. (2013) and Wecht et al. (2014) (in situ estimates and GOSAT inv) used meteorological models to estimate fluxes, similar to the work presented here. Tracer-tracer relations were used for the TCCON (Wunch et al., 2009, 2016), CLARS (Wong et al., 2015, 2016), and in situ observations (Hsu et al., 2010; Wennberg et al., 2012). The "W09 adj" are the results of Wunch et al. (2009) when adjusted for our posterior $CO_2$ flux. Methane in situ results for CalNex (May–June 2010) are from Wennberg et al. (2012). CO in situ results in 2002 and 2010 are from Brioude et al. (2013).

model above a threshold. We start with a threshold that is a factor of 64 and adjust from there. We also adjust values of $\mathbf{x}_a$, $\mathbf{S}_\epsilon$, and $\mathbf{S}_a$ by factors of $2^{-10}$ to $2^{10}$. These results show the overall flux generally has low sensitivity to scaling $\mathbf{S}_\epsilon$, and $\mathbf{S}_a$ but has some sensitivity when filtering more data and about a 10 % sensitivity to the scaling of $\mathbf{x}_a$. The interannual variability, which we expect is less than about 25 %, increases for large $\mathbf{S}_a$. Increasing $\mathbf{S}_a$ increases $r$, and the degrees of freedom for the signal

5 ($\mathrm{dof_s}$) with only a small effect on the overall flux, but also increases the interannual range. Decreasing $\mathbf{S}_\epsilon$ increases $r$ and $\mathrm{dof_s}$, but it also increases $\chi^2$ and the interannual range. We estimate an overall uncertainty of 10 % from these parameters.

Hedelius et al. (2017b) reported $2\sigma$ measurement bias of less than $\sim$0.2 ppm $\mathrm{X_{CO_2}}$ (central estimate, maximum range $<$0.5 ppm) between the AFRC and Caltech TCCON sites, but even a bias of 0.2–0.3 ppm $\mathrm{X_{CO_2}}$ will produce an error of $\sim$10 % in the flux. This bias could also arise from improper boundary conditions or application of averaging kernels.

10 We test the sensitivity to different inversion and modeling schemes (Table 3). The Kalman filter (Appendix D) and the non-linear inversion (Eq. E2) results are not unreasonable for $CO_2$. However, their $CH_4$ flux results are unreasonably low, likely



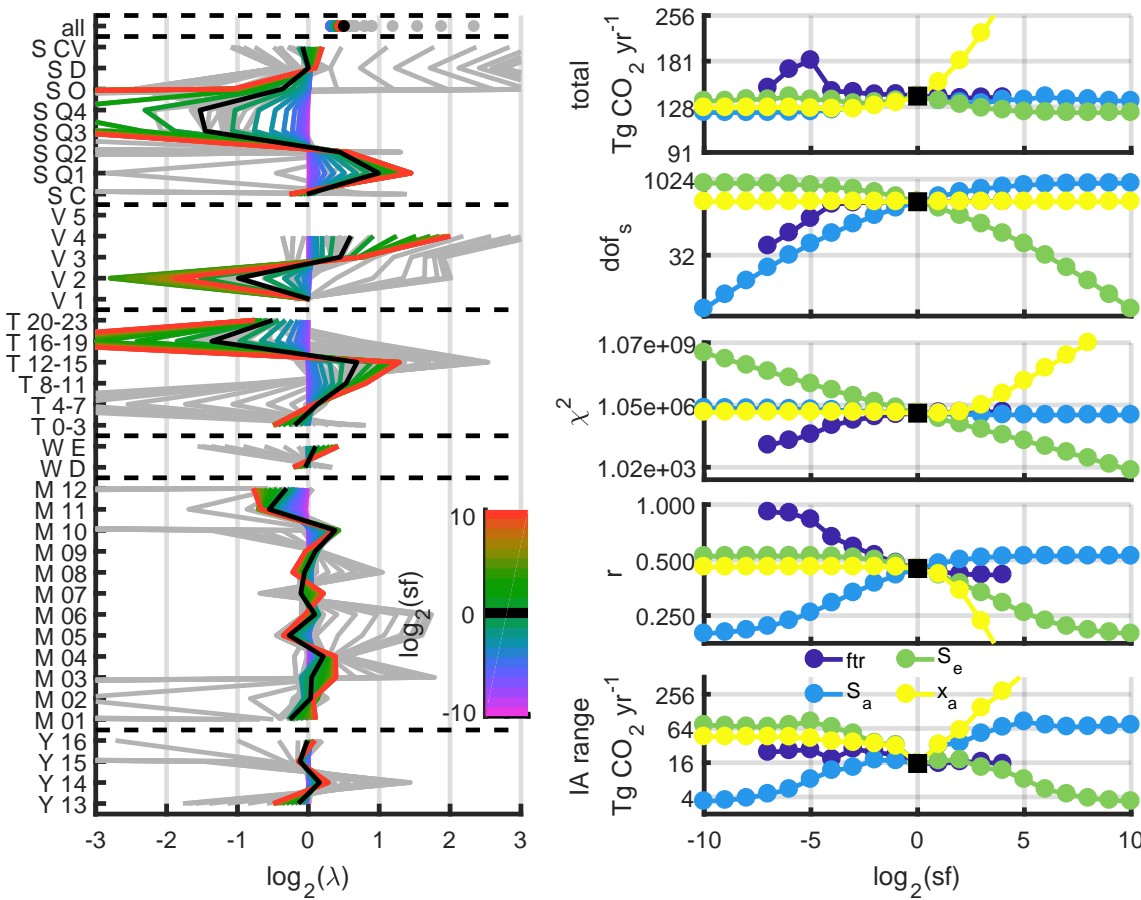

**Figure 7.** Assessment of sensitivity to initial values. On the left is the reduced state vector with 7 categories (overall, spatial, vertical, time of day, weekday/end, month, and year). $\lambda$ values indicate how much the prior is scaled on average compared to other elements in its category. Gray lines are results from all tests, and colored lines are from the $\mathbf{S}_a$ test. As $\mathbf{S}_a$ gets larger, the variability in the retrieved $\gamma$ factors increases. The right shows overall fit parameters, including the overall flux, the $\mathrm{dof}_s$, $\chi^2$, Pearson's correlation coefficient between observed and post-inversion model values, and the interannual range. Note the $\log_2$ axes which indicate the magnitude of change in the sensitivity test compared with the base case (sf=scaling factor). Moving left filtering (ftr) becomes more stringent, constraints on $\mathbf{S}_a$ or $\mathbf{S}_e$ are increased, or $\mathbf{x}_a$ is scaled down. Generally the total flux is unchanged except for scaling $\mathbf{x}_a$ which increases the flux by about 10 % of the change in the prior. Here the goal was to simultaneously increase $\mathrm{dof}_s$, decrease $\chi^2$, and increase $r$ while keeping the interannual variability below about 25 %.

from high model:measured values having unreasonably high weights in these particular schemes with few scaling factors (Appendix D2). GDAS and HRRR results are within uncertainty.

In summary, we estimate a 20 % uncertainty from model winds, 10 % from our choice of initial values, 10 % from observations and the boundary condition, 5 % from the prior flux (based on results of Lauvaux et al., 2016), and 5 % from additional





**Table 3.** Fluxes from various methods.

| Method | Tg $CO_2$ yr$^{-1}$ | Gg $CH_4$ yr$^{-1}$ | Gg CO yr$^{-1}$ |
|---|---|---|---|
| GDAS (0.5°)[a] | 104±26 | 365±91 | 530±133 |
| HRRR (3 km) | 147±37 | 439±110 | 616±154 |
| NAM (12 km) | 139±35 | 325±81 | 555±139 |
| Kalman filter | 97±24 | 104±26 | 379±95 |
| non-linear inv. | 151±38 | 103±26 | 374±94 |

For a given gas, all the inversions use the same observed $\Delta X_{gas}$ (Caltech TCCON − AFRC TCCON). The top 3 rows are from using different meteorological models, with the same inversion scheme (Eq. E3). The last 3 rows are from using the same meteorological model (NAM 12 km) with different inversion schemes. Errors are 25 %. [a]For GDAS $\mathbf{S}_a$ is 30× smaller

random uncertainty. The sum in quadrature is 25 %. By comparison, uncertainty estimates from other inversions were 11 % (inner 50 percentile range from an ensemble) for Indianapolis (Lauvaux et al., 2016), and 5 % for the Bay Area using pseudo-observations (Turner et al., 2016). Both of these studies benefited from additional sites (9 and 34 respectively) and custom WRF model runs. Ye et al. (2017) estimated an uncertainty of 5 % for the SoCAB flux by using data from 10 OCO-2 tracks,
however this is not directly comparable with our result because it does not include uncertainty from biases in the forward model, observations, and inversion scheme.

## 5    Discussion

### 5.1    Emission Ratios

Emission ratios can help us evaluate the inversion for the SoCAB. Previous studies (Newman et al., 2016; Wunch et al., 2009,
2016) have noted that the Pasadena area is a good receptor site for the basin, so tracer-tracer ratios observed there should approximately correlate with emission ratios. If the ratios are significantly different it could highlight an error in the inversion scheme, or the a priori assumption of sources. However, errors in the model can be correlated for different tracers which would obscure universal biases to all gases. Interannual ranges for CO, $CH_4$, and $CO_2$ are 19 %, 13 %, and 11 % compared to their central estimates. The interannual range of ratios for CO:$CH_4$, and CO:$CO_2$ are similar at 21 % and 22 % respectively, but the
$CH_4$:$CO_2$ range is smaller at 2 %.

We estimate emission ratios using the solar zenith angle (SZA) anomaly method described by Wunch et al. (2009, 2016), as well as from the average enhancement compared with AFRC or the Pasadena:Lancaster gradient ratio. Errors are assumed to equal the standard deviation of all the data, and a linear fit is made using the methods of York et al. (2004). We estimate the emission ratio from the work of Verhulst et al. (2017) using the weighted mean of the excess ratios from their 5 in-basin
sites, with weights $\frac{1}{\sigma^2}$. Emission ratios from the SZA anomaly method and the differenced enhancement are in agreement with ratios from previous studies (Fig. 8). The $CH_4$:$CO_2$ from the inversion is slightly lower than but similar to other studies. The




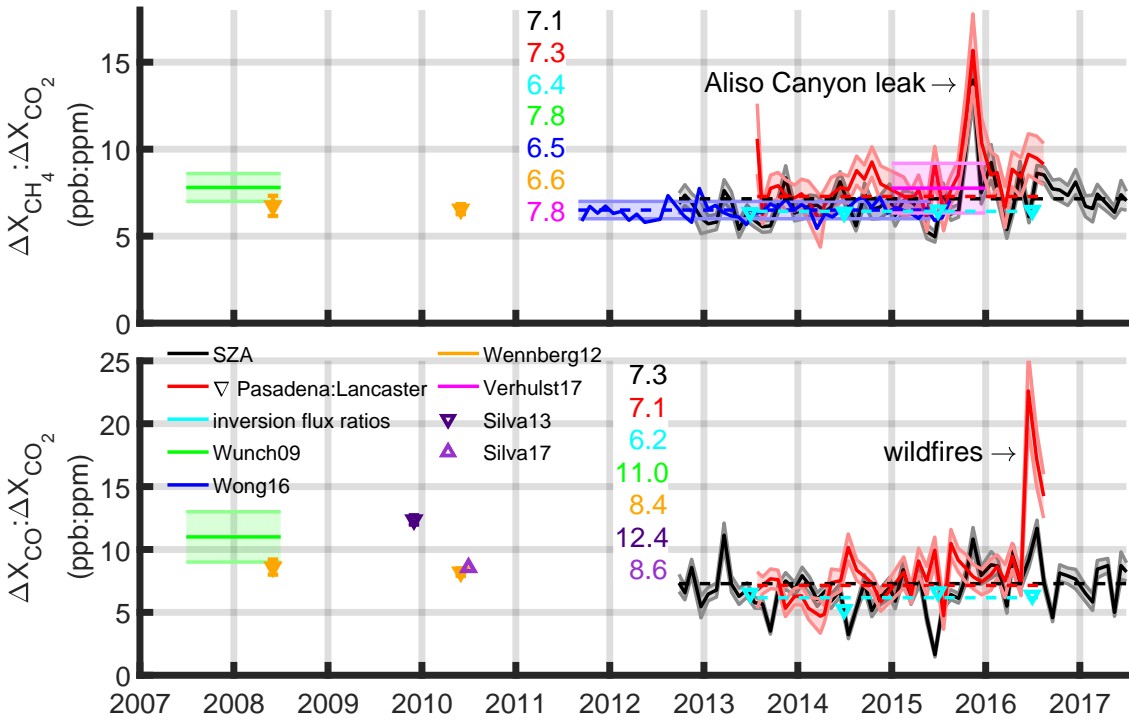

**Figure 8.** Emission ratios compared with previous studies. Numbers shown are central values from the different methods and studies. Overall fits are shown as dashed lines. The ratio from the Pasadena (Caltech) to Lancaster (AFRC) gradients is based on the difference between the TCCON observations. Values from Silva et al. (2013); Silva and Arellano (2017) were part of global studies.

$CO:CO_2$ ratio is also lower. Based on the CARB inventories, a decrease is expected because CO emissions have decreased more than $CO_2$ emissions over the past decade.

In November 2015, the large $CH_4:CO_2$ ratio is from additional methane emissions from the Aliso Canyon gas leak (Conley et al., 2016). Though this leak persisted until February 2016, different wind patterns caused less of the highly methane enriched air to be transported and observed in Pasadena after the first 2 months. The large $CO:CO_2$ ratios seen in summer 2016 are from wildfires. The San Gabriel Complex Fire was less than 25 km to the east and burned 22 km² over a month. It was close enough for ash to be transported to Pasadena. Eight other major fires within 150 km burned an additional 400 km² during June–August 2016 (https://firetracker.scpr.org/wildfires/archives/).

### 5.2 Weekend effect

The weekday to weekend flux ratios (WD:WE) are listed in Table 4. The uncertainty is estimated to be ±0.17 based on changes in the ratio from the $\mathbf{S}_a$ scaling test up to $8\times$ (Fig. 7). Weekday:weekend ratios are larger than those from previous studies for CO (Pollack et al., 2012; Brioude et al., 2013). Compared with the prior, the $CO_2$ ratio is scaled down, and the CO ratio remains equal to the prior. Methane has a ratio that is larger than unity. In contrast to our inversion results, methane is not expected



**Table 4.** Weekday:weekend emission ratios.

|  | $CO_2$ | CO | $CH_4$ |
|---|---|---|---|
| Pollack et al. (2012)[a] | $1.10 \pm 0.32$ | $1.08 \pm 0.31$ |  |
| Brioude et al. (2013)[b] | $0.91 \pm 0.09$ | $1.17 \pm 0.19$ |  |
| TIMES[c] | 1.09 |  |  |
| Hestia-LA v1.0[d] | 1.23 |  |  |
| This study | $1.11 \pm 0.17$ | $1.23 \pm 0.17$ | $1.17 \pm 0.17$ |

[a]WD:WE CO ratios from Pollack et al. (2012) were calculated using the difference between the CalNex-Pasadena and Mt. Wilson Flasks (Table 2 therein). For $CO_2$ we used the CO WD:WE ratios with the $CO/CO_2$ WD:WE ratios in Pasadena (Table 3 therein). [b]WD:WE ratios from Brioude et al. (2013) were calculated by assuming ratios between daytime and all day emissions in the posterior were equal using Table 3 therein. [c]Temporal Improvements for Modeling Emissions by Scaling (TIMES) were reported for the contiguous United States by Nassar et al. (2013). [d]Hestia-LA is based on Fig. 2 from Hedelius et al. (2017a). This same ratio is used in the CO and $CO_2$ priors in this study.

to vary significantly on weekdays compared to weekends because production from biogenic sources and fugitive losses from natural gas infrastructure are less time variant than CO and $CO_2$ emissions.

## 6 Conclusions

This study demonstrates a method to readily obtain estimates of net $CO_2$ fluxes over regions on order of 10,000 km using only remote sensing observations. This method could be applied almost anywhere globally using only OCO-2 or other space-based observations of $CO_2$ (e.g., GOSAT) without the need for ground observations, or a specialized model. Our estimates of total annual $CO_2$ fluxes from the SoCAB using HYSPLIT with NAM 12 km as our dynamical model are similar to some previous estimates (Fig. 4), but less than inventory values reported in tracer-tracer flux estimate papers (Wunch et al., 2009; Wong et al., 2015). This has important implications for these studies, which would have overestimated $CH_4$ emissions if $CO_2$ emissions were also too large. Net CO and $CH_4$ fluxes are slightly less than previous studies.

The overall uncertainty is 25 %, with the dynamical model contributing the most. We consider an uncertainty of 25 % to be large and shows additional work is needed to improve constraints. If errors are from persistent biases, then relative changes in time can be observed, though such changes might also be observed using just the observations without a model (e.g., Kort et al., 2012). The wide range of uncertainty suggests that $CO_2$ flux estimates from the SoCAB will benefit from additional measurements—such as the LA Megacity Carbon Project in situ tower network (Verhulst et al., 2017), the planned geostationary GeoCARB mission, and the OCO-3 mission which has a raster mode that can scan throughout the basin. Further improvements in modeling and inversion techniques will also help, including assimilating all available observations (in situ network, TCCON, CLARS, OCO-2, and GOSAT). These additional surface and space-based observations can aid in not only improving the accuracy of the overall flux, but also may be incorporated into spatiotemporal inversions to map fluxes from sub-




regions of the SoCAB with confidence. Understanding the contributions from the biosphere will also be important to diagnose how much carbon is from fossil fuels. For example, Newman et al. (2016) showed the typical contribution of the biosphere to the excess $CO_2$ in Pasadena was 8–16 % as large as the fossil fuel contribution using $\Delta^{14}C$ observations.

*Data availability.* TCCON data used in this study (GGG2014) are hosted on the TCCON data archive (https://tccondata.org/) and are used in
accordance with the Data Use Policy (https://tccon-wiki.caltech.edu/Network_Policy/Data_Use_Policy). OCO-2 data are hosted by Goddard Earth Sciences (GES) Data and Information Services Center (DISC) (https://disc.gsfc.nasa.gov/datasets/OCO2_L2_Lite_FP_8r/summary). ODIAC2016 data are hosted by NIES (http://db.cger.nies.go.jp/dataset/ODIAC/). Nightlight products were obtained from the Earth Observation Group, NOAA National Geophysical Data Center and are based on Suomi NPP satellite observations (http://ngdc.noaa.gov/eog/viirs/). Gridded Harvard-EPA emissions are hosted on the EPA website (https://www.epa.gov/ghgemissions/gridded-2012-methane-emissions).
NOAA gridded meteorological data are hosted on the NOAA ARL server (https://www.ready.noaa.gov/archives.php).

The CARB regularly publishes emission inventories of various gases. CO inventories are available online (2017: https://www.arb.ca. gov/app/emsinv/2017/emssumcat.php, 2013: https://www.arb.ca.gov/app/emsinv/2013/emssumcat.php, 2009: https://www.arb.ca.gov/app/ emsinv/fcemssumcat2009.php), as are $CH_4$ inventories (2017: https://www.arb.ca.gov/app/ghg/2000_2015/ghg_sector_data.php, 2013: https: //www.arb.ca.gov/app/ghg/2000_2011/ghg_sector_data.php, 2009: https://www.arb.ca.gov/app/ghg/2000_2006/ghg_sector.php).

## Appendix A: Observation data filtering, and boundary condition

GOSAT-ACOS v2.9 $X_{CO_2}$ levels are enhanced by only $3.2 \pm 1.5 \,(1\sigma)$ ppm in the SoCAB (Kort et al., 2012). This means a bias of 0.3 ppm could lead to a 10% bias in the flux. Thus it is critical to account for biases down to the tenths of a ppm level or better. This is a challenge given that the accuracy of OCO-2 (v7r) over land had been estimated as 0.65 ppm (Worden et al., 2017), and OCO-2 comparisons with TCCON range from -0.1 ppm to 1.6 ppm (Wunch et al., 2017).

## A1   Quality filters

Compared with the TCCON, OCO-2 spectra are lower resolution. OCO-2 observations are also sensitive to surface albedo, and are more sensitive to aerosol scattering than solar-viewing instruments. These sensitivities can cause spurious results which need to be filtered out. Included in the OCO-2 data is a binary flag as well as warn levels (WL) for quality filtering, where higher WLs indicate less reliable data. WL definitions are different for v7 and v8, but here we use the binary $X_{CO_2}$ filter and
only include v8 data with a WL$\leq 1$. WL$\geq 4$ data are already removed by the binary flag. This leaves 2,714 observations.

For TCCON observations we use the public data, which already has some static within-range filters applied. We also exclude data that differ from the model by a factor of 64 or greater, leaving 5,060 observations.

## A2   Background, boundary conditions, and averaging kernels

To eliminate the ambient $X_{gas}$ levels that would be observed in the absence of local emissions, we subtract values measured
by the AFRC TCCON site from both the Caltech TCCON and OCO-2 data obtained in the basin. We choose TCCON data





as background for OCO-2 to reduce the likelihood of albedo related bias from using OCO-2 observations over the Mojave desert (Wunch et al., 2017) as well as the chance of inducing a bias from using different viewing modes by using ocean glint observations. For expanding these methods globally, OCO-2 observations not directly influenced by the source could be used as background. For example, Janardanan et al. (2016) categorized space-based observations of $X_{CO_2}$ by making a forward

model estimate of $X_{CO_2}$ enhancements from fossil fuel combustion and setting a threshold to define as polluted or unpolluted. Such an approach could work globally, but may have errors if there are errors in the prior emissions or transport model.

Because we expect most of the difference to arise from polluted air near the surface, we divide the enhancements by the surface averaging kernels of the in-basin observations. OCO-2 surface averaging kernels in the basin are $0.986\pm0.010\,(1\sigma)$ with a 99 % confidence interval of 0.955 to 1.016. TCCON surface averaging kernels depend on surface pressure and solar

zenith angle (SZA) and are $0.96\pm0.14\,(1\sigma)$ throughout the full range of observations.

Even in the absence of local anthropogenic emissions the $X_{CO_2}$ measured within the SoCAB could be different from that measured outside by a few tenths of a ppm because of different measurement heights and atmospheric $CO_2$ profiles. We account for a boundary condition of the form:

$$b_{\text{gas}} = \left( \frac{X_{\text{gas,a,S}}}{X_{\text{gas,a,B}}} - 1 \right) \hat{X}_{\text{gas,B}}, \tag{A1}$$

where subscript $a$ represents the a priori, S represents a measurement within the SoCAB, B represents the background, and the hat represents a retrieved value. Equation A1 can be interpreted as the difference that would be observed between sites due to differences in the gas vertical profiles. The a priori profiles do not include local anthropogenic emissions. The boundary condition is subtracted from the SoCAB−AFRC difference.

## Appendix B: Dynamical model error

The dynamical model could have errors in the PBL height estimation as well as in the wind speed and direction. In a case study for spring 2011 and 2012 primarily over the Midwestern U.S. a NAM temperature derived PBL height had a mean bias of about $-50\,$m, with an inner 50 percentile range of about $\pm250\,$m (Coniglio et al., 2013). For wind error we compare with 10 m winds from the San Gabriel (El Monte) Airport 10 km SE of Caltech (34.083°, -118.033°, 90 m a.s.l.). We assume the winds are the same at both locations. Airport meteorological data are obtain through the NOAA National Centers for Environmental

Information (https://www.ncdc.noaa.gov/cdo-web/datatools/lcd).

Trajectory speed and direction are estimated based on when and where trajectories ending at 50 m a.g.l. enter a 5 km radius circle around the receptor site. Results are shown in Fig. B1. The mean speed of HYSPLIT trajectories is less than what is expected by comparing with the surface winds. In contrast, previous studies have shown high model wind speed bias near the surface at the LAX airport, 34 km SW and near the coast (Feng et al., 2016; Angevine et al., 2012; Ye et al., 2017). The

difference biases could in part be from coastal versus inland however, Feng et al. (2016) also showed a high model bias closer to Caltech. Model differences, the 10 km horizontal and $\sim150\,$m height difference between Caltech and the airport could also



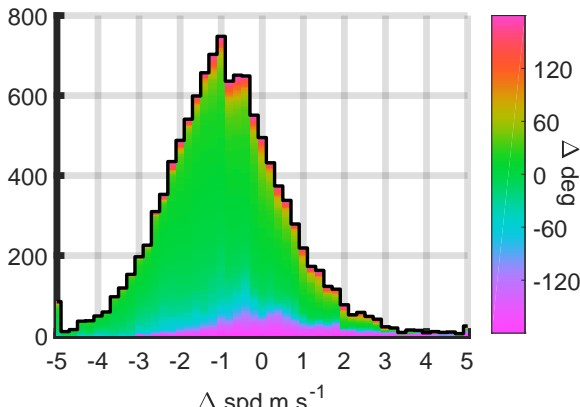

**Figure B1.** Histogram of Wind speed errors (HYSPLIT-measured) compared to surface observations at the San Gabriel airport. The mean error is $-0.9\,\mathrm{m\,s^{-1}}$ ($-30\,\%$), the 95 % confidence interval is [-3.7, 2.3] $\mathrm{m\,s^{-1}}$. The mean direction error is less than $5°$, and 75 % of direction errors are within $\pm 45°$.

contribute to the discrepancy. We expect the average bias throughout the PBL to be lower than at the surface, and assign an uncertainty of up to $\sim 20\,\%$ to the average wind.

## Appendix C:  Residence times from HYSPLIT

HYSPLIT mean trajectories are air parcel locations at different heights for select times (in our case, every 20 min). These are
aggregated and normalized for each $0.01° \times 0.01°$ pixel and for each hour. First each trajectory is interpolated to 1 s positions. Then we determine the vertical fraction of the mixing layer (ML) the trajectory takes. This fraction is the vertical spacing between trajectories (in hPa) divided by the ML depth (in hPa). The HYSPLIT mixing depth is based on the underlying Eulerian model. Parcels above the ML get counted as zero. Then we count how long any parcel was in each pixel (in s) to get the residence time. Monthly average examples of this are shown in Fig. C1. The residence time is multiplied by the a priori
flux to determine the column enhancement (in $\mathrm{g\,m^{-2}}$). By dividing by a model estimate of the dry-air $\mathrm{mol\,m^{-2}}$ based on model surface pressure we obtain a forward model estimate of the $\mathrm{X_{gas}}$ enhancement from local sources (in ppm or ppb).

## Appendix D:  Kalman filter

The Kalman filter used to estimate SoCAB $CO_2$ emissions is based on methods described by Kleiman and Prinn (2000) with modifications. This is an iterative approach using a single overall scaling factor. The difference in $\mathrm{X_{CO_2}}$ between measurements
in the SoCAB and AFRC is the observed measurement, $y^{\mathrm{obs}}$. The error $\sigma_y$ associated with each $y^{\mathrm{obs}}$ is estimated from the sum





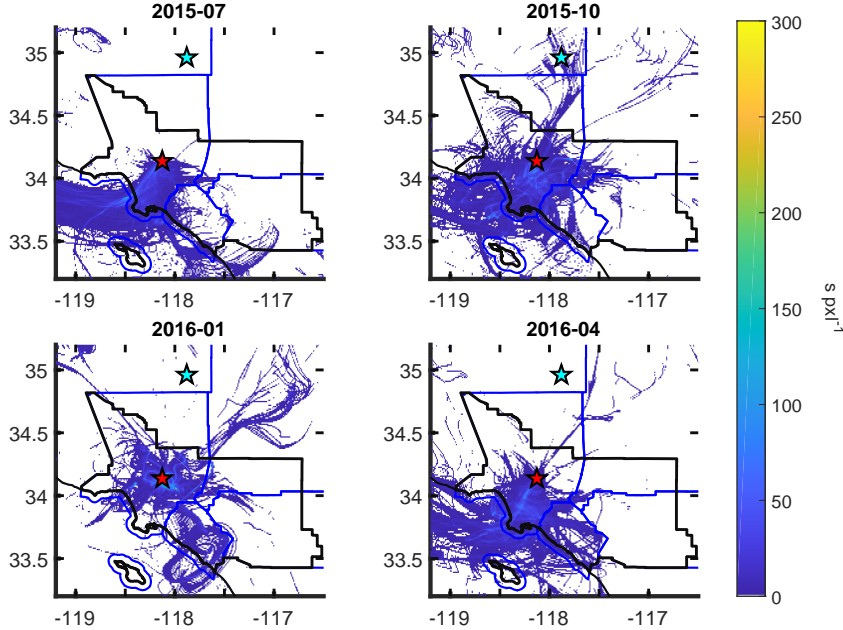

**Figure C1.** Maps of monthly averaged residence times in the ML per pixel for trajectories ending at 21 UTC, shown for all times leading up to the observation. Pixels are $0.01° \times 0.01°$, or approximately $1.03\,\mathrm{km}^2$. In July the origins were more predictable, but in January there was greater variation.

in quadrature of the error from each site, i.e.,

$$\sigma_y = \sqrt{\bar{\hat{y}}_{\mathrm{err,C}}^2 + \bar{\hat{y}}_{\mathrm{err,A}}^2}, \tag{D1}$$

where the subscript C is for Caltech (or measurements in the SoCAB), and A is for AFRC (or 'background'). The error of the averaged data for an individual site is calculated by

$$5 \quad \bar{\hat{y}}_{\mathrm{err}} = \sqrt{\frac{1}{n-1} \frac{1}{\sum \hat{z}_{i,\mathrm{err}}^{-2}} \sum_i \hat{z}_{i,\mathrm{err}}^{-2} \left(\hat{z}_i - \bar{\hat{y}}\right)^2}, \tag{D2}$$

where $n$ is the number of measurements, $\hat{z}_i$ are the individual $X_{CO_2}$ measurements, $\hat{z}_{i,\mathrm{err}}$ are the reported errors associated with the measurements, and $\bar{\hat{y}}$ is the weighted average using $\hat{z}_{\mathrm{err}}^{-2}$ as weights. Note that Eq. D2 takes into account both the measurement errors as well as the spread of the measurements.

### D1 Iterations

10 We initialize the iterations with an arbitrary scaling factor $\alpha_0 = 1$ and an associated error of $\sigma_{\alpha 0} = 0.7$. These initial values have little influence on the final result.



We iterate over the $k$ measurements by calculating the partial derivative:

$$h_k = \frac{\partial y_k^{\text{est}}}{\partial \alpha_k} = \sum_j s_{j,k} t_{j,k}, \tag{D3}$$

where subscript $j$ is for a particular grid box, $s$ is the a priori surface flux, and $t$ is the residence time. Equation D3 is identical to Eq. A2 in Kleiman and Prinn (2000). Because this is a scaling retrieval, $h_k$ is the observation operator. We can multiply it

by the state element ($\alpha$) to obtain the estimated observation (Eq. D6). The gain scalar $g_k$ and new state error are calculated by (Eq. A4 and A5 in Kleiman and Prinn (2000)):

$$g_k = \sigma_{\alpha,k-1}^2 h_k \left( h_k^2 \sigma_{\alpha,k-1}^2 + \sigma_{z,k}^2 \right)^{-1}, \tag{D4}$$

$$\sigma_{\alpha,k}^2 = \sigma_{\alpha,k-1}^2 \left( 1 - h_k g_k \right). \tag{D5}$$

We make a modification to calculate the estimated measurement, omitting the term for the convergence of fluxes due to unresolved motions in the transport model. The estimated forward model is

$$y_k^{\text{est}} = \alpha_{k-1} h_k, \tag{D6}$$

and the state estimate is

$$\alpha_k = \alpha_{k-1} + g_k \left( y_k^{\text{obs}} - y_k^{\text{est}} \right). \tag{D7}$$

**D2    A note on single scale factor inversions with large outliers**

Some single scale factor inversions can be written in the form

$$\hat{\mathbf{y}} = \lambda \mathbf{y}_{\text{mod}}, \tag{D8}$$

where "mod" represents the initial model values. We consider the case when cost function is of the form:

$$\mathbf{J}_c = \frac{(\mathbf{y}_{\text{obs}} - \lambda \mathbf{y}_{\text{mod}})^2}{\mathbf{y}_{\text{err}}^2} \tag{D9}$$

where the error term accounts for both the model and observation errors. If the error is not a function of $\lambda$ then

$$\frac{\partial \mathbf{J}_c}{\partial \lambda} = 2 \frac{(\lambda \mathbf{y}_{\text{mod}}^2 - \mathbf{y}_{\text{obs}} \mathbf{y}_{\text{mod}})}{\mathbf{y}_{\text{err}}^2}. \tag{D10}$$

Setting Eq. D10 equal to zero and solving for $\lambda$ yields

$$\lambda = \frac{\Sigma_k y_{\text{obs},k} y_{\text{mod},k}}{\Sigma_i y_{\text{mod},k}^2}. \tag{D11}$$

Note the change from vector to summation notation. Equation D11 is a first order estimate of the overall scale factor $\lambda$. This

indicates that $\lambda$ can be low with high model:observation ratios which heavily weight the result.



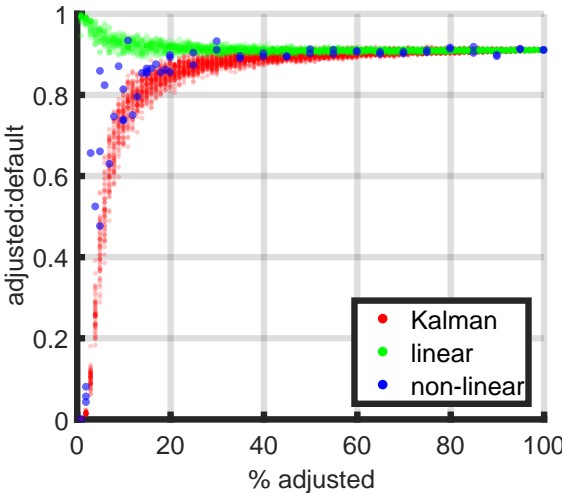

**Figure D1.** Effects of scaling a random subset of model data compared to no scaling. When fewer points are scaled, they are scaled by a larger amount. The Kalman filter and non-linear inversion are more affect by a few strong outliers than the linear inversion.

This is demonstrated in a sensitivity test, where we scale a subset of points (Fig. D1). We create pseudo-observed values by using the original model values. We create pseudo-model data by scaling a random subset of the original model data by $(1.1)^{\frac{n}{s}}$ where $n$ is the total number of points, and $s$ is the number in the subset. For example, when $100\,\%$ of the model points are adjusted we scale them all up by $10\,\%$. The test is repeated multiple times, with fewer repeats for the non-linear model because

it takes the longest. These results show that having a few large outliers in the Kalman filter (1 scale factor) and the non-linear (40 scale factor, Sect. E) inversions can significantly pull the results compared with the linear ($\sim$1,000 scale factor, Sect. E) inversion.

### Appendix E:  Bayesian inversions

The Bayesian approach to solving atmospheric inverse problems has been described in more detail by Rodgers (2000) (see
Section 2.3.2). Turner et al. (2016) describes this approach for an urban region. Here we follow the notation of Rodgers (2000). For scaling retrievals, Bayesian inversions minimizes a cost function ($-2\ln P(\mathbf{y}|\mathbf{x})$) of the form in Eq. D9. This assumes error statistics are adequately known, and are Gaussian for both the state vector $\mathbf{x}$ (length $n$) and the measurement vector $\mathbf{y}$ (length $m$).

### E1    Forward model

The generalized forward model can be written as

$$\mathbf{y} = \mathbf{F}(\mathbf{x}) + \boldsymbol{\epsilon}, \tag{E1}$$





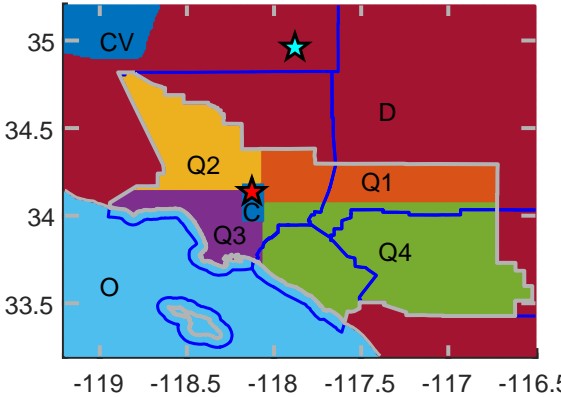

**Figure E1.** Extent of the 8 spatial sub-regions. C=center, Q1–Q4 are SoCAB quadrants, O=ocean, CV=Central Valley, D=all other areas, mostly the Mojave Desert to the northeast.

where $\epsilon$ is an error term. We test 2 similar forward models for the Bayesian inversions. The first is chosen to reduce the number of elements in the state vector. This choice was made based on having only 2 measurement locations. This model has 40 state vector scaling factors $\lambda$ in 7 different classes corresponding to year (6), month (12), weekday/end (2), time of day (6), vertical level (5), spatial bin (8), and overall (1). Time of day bins cover 4 hours each with local ending times at 3:00, 7:00, 11:00, 15:00, 19:00, and 23:00. Aggregated vertical bins are each about 3.5 % of the atmosphere, split at 300, 612, 936, 1272, and 3200 m agl. These are designed to help diagnose transport or footprint extent errors, and the upper 2 levels are weighted less when estimating the total SoCAB flux. Spatial bins (Fig. E1) were chosen with one over the ocean, one over Central Valley, one for the rest of the area outside the SoCAB, and five inside the SoCAB. Each SoCAB area has approximately the same influence on observations at the Caltech (abbreviated CIT) site based on residence times. This model is

$$\mathbf{F}\left(\mathbf{x}\right) = \lambda_{all} \sum_{yr=1}^{6} \sum_{mth=1}^{12} \sum_{dow=1}^{2} \sum_{tod=1}^{6} \sum_{vbin=1}^{5} \sum_{sbin=1}^{8} \lambda_{yr}\lambda_{mth}\lambda_{dow}\lambda_{tod}\lambda_{vbin}\lambda_{sbin} \left(m_{j,\mathrm{CIT}} - m_{j,\mathrm{AFRC}}\right). \tag{E2}$$

Here, $m = \Sigma t \times s$ is the model amount determined by multiplying the residence time $t$ by the a priori surface flux $s$ and summing over all times and $0.01° \times 0.01°$ grid boxes in the bin. We use $j$ here as shorthand for the subscript "$yr, mth, dow, tod, vbin, sbin$."

We also use a similar linear model of the form

$$\mathbf{F}\left(\mathbf{x}\right) = \sum_{j=1}^{34,560} \lambda_j \left(m_{j,\mathrm{CIT}} - m_{j,\mathrm{AFRC}}\right). \tag{E3}$$

In this form there are up to nearly 35,000 original elements in our state vector as opposed to the 40 elements in Eq. E2. Most of the original elements are not ever sampled (e.g., from 2012 and 2017) and not used when reporting our total fluxes. We remove elements which are not linearly independent which reduces the actual number used to less than (about one-fifth) the number of



observations. We select the most important elements from matrix $\mathbf{R}$ found by performing a QR decomposition on the $\mathbf{K}$ matrix. Changing the cutoff (and $\mathbf{S}_a$) affects the sensitivity to the prior.

**E2 Solutions**

For the linear forward model (Eq. E3), the retrieved state vector ($\hat{\mathbf{x}}$) can be found in a single step,

$$\hat{\mathbf{x}} = \mathbf{x}_a + \mathbf{S}_a \mathbf{K}^T \left( \mathbf{K} \mathbf{S}_a \mathbf{K}^T + \mathbf{S}_\epsilon \right)^{-1} (\mathbf{y} - \mathbf{K} \mathbf{x}_a). \tag{E4}$$

$\mathbf{x}_a$ denotes the a priori state vector. $\mathbf{S}_a$ is the a priori covariance matrix for the state vector (denoted $\mathbf{B}$ in some texts). $\mathbf{K}$ is the $m \times n$ Jacobian matrix (denoted $\mathbf{H}$ in some texts). $\mathbf{S}_\epsilon$ is the $m \times m$ measurement error covariance matrix (denoted $\mathbf{R}$ in some texts), which includes errors from both the observations and the forward model. $\mathbf{S}_\epsilon$ is often treated as a diagonal matrix, with $\sigma_k^2$ values along the diagonal.

For a non-linear forward model (e.g., Eq. E2), the inverse solution can be found using an iterative Levenberg-Marquardt method. This is described in more detail by Rodgers (2000) Section 5.7. The iterative solution is:

$$\mathbf{x}_{i+1} = \mathbf{x}_i + \left[ (1+\gamma) \mathbf{S}_a^{-1} + \mathbf{K}_i^T \mathbf{S}_\epsilon^{-1} \mathbf{K}_i \right]^{-1} \left\{ \mathbf{K}_i^T \mathbf{S}_\epsilon^{-1} \left[ \mathbf{y} - \mathbf{F}(\mathbf{x}_i) \right] - \mathbf{S}_a^{-1} \left[ \mathbf{x}_i - \mathbf{x}_a \right] \right\}. \tag{E5}$$

The symbol $\gamma$ is a factor chosen at each iteration to minimize the cost function based on how $\chi^2$ changes, and $i+1$ denotes the current iteration.

**E3 A priori values**

We define values for $\mathbf{x}_a$, $\mathbf{S}_\epsilon$, and $\mathbf{S}_a$. First, our state vector is composed of scaling factors and all elements in $\mathbf{x}_a$ are unity for CO and $CH_4$. Because ODIAC2016 emissions for the SoCAB are low compared to other inventories we use 1.25 for $CO_2$. $\mathbf{S}_\epsilon$ is a diagonal matrix. Along diagonal elements are the errors from the observations plus the errors from the transport model. Observation errors are $\sigma_y^2$ determined from Eq. D1. We assume transport errors are constant and equal to the overall median observation error.

For simplicity, $\mathbf{S}_a$ is chosen as a single scalar value for the linear model (Eq. E3). We select $\mathbf{S}_a$ values which keep the interannual variability under about 25 %, and minimize dependence on the prior as noted by a sensitivity test scaling the prior (Fig. 7). This is also a trade-off between maximizing the degrees of freedom and $r$, avoiding unstable conditions, and minimizing $\chi^2$. $\mathbf{S}_a$ is tuned to 0.07 for $CO_2$, 0.007 for $CH_4$, 0.002 for CO, and 0.2 for $CO_2$ using OCO-2 observations. Generally as $\mathbf{S}_a$ increases the interannual range increases, but the dependence on the prior decreases. These values were selected to have the smallest dependence on the prior while keeping the interannual range within our arbitrary 25 % limit. For the 40 factor inversion looser constraints are used with diagonal values of $CO_2$: 0.7, $CH_4$: 0.7, CO: 0.04, and $CO_2$ (OCO-2): 7. Off-diagonal values between adjacent elements (e.g., years, months) are one-third of those along the diagonal in the 40 factor inversion, which is a somewhat arbitrary choice based on our a priori guess of how strongly adjacent elements are related.



*Competing interests.* The authors declare that they have no competing interests.

*Acknowledgements.* The authors wish to acknowledge providers of data. OCO-2 lite files were produced by the OCO-2 project at the Jet Propulsion Laboratory, California Institute of Technology. Resources supporting OCO-2 retrievals were provided by the NASA High-End Computing (HEC) Program through the NASA Advanced Supercomputing (NAS) Division at Ames Research Center. Nightlight products

5 were obtained from the Earth Observation Group, NOAA National Geophysical Data Center and are based on Suomi NPP satellite observations. The $0.1°$ methane inventory was produced by Harvard University in collaboration with the EPA. The authors gratefully acknowledge the NOAA Air Resources Laboratory (ARL) for the provision of the HYSPLIT transport model (http://www.ready.noaa.gov) as well as the gridded archived meteorological data used in this publication.

We thank Ron Cohen, Nick Parazoo, Anna Karion, and Taylor Jones for helpful discussions.

10 This work was financially supported by NASA's OCO-2 project (grant no. NNN12AA01C), and NASA's carbon cycle and ecosystems research program (grant no. NNX14AI60G and NNX17AE15G). Tomohiro Oda is supported by the NASA Carbon Cycle Science program (grant no. NNX14AM76G).



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
