# Peer review of "Southern California Megacity CO2, CH4, and CO flux estimates using ground and space-based remote sensing and a Lagrangian model"

_Atmospheric Chemistry and Physics, 2018_

## Referee Comment (RC1) · Anonymous Referee #1 · 27 Jun 2018

General Comments

Hedelius et al present flux estimates of CO2, CH4, and CO from Southern California using an inversion that combines satellite- and surface-based observations, with the HYSPLIT model. The paper is well written, within the scope of ACP, and provides substantial technical detail for the reproducibility of their results. I recommend accepting this work for publication after the following issues are addressed.

My main concern is regarding the a priori flux estimates used in this work, particularly given the acknowledged existing higher accuracy inventories. I understand that the methodology was designed to be applicable globally, but it is not clear how much the

quality of the inversion suffers from this goal.

How different are the CO2 fluxes using the modified ODIAC as compared to using Hestia-LA? Lauvaux et al (2016) used a different Hestia data product and tower measurements in a substantially smaller city; it isn't obvious that the comparison holds over SoCAB with remote sensing data.

Given a lack of information on landfills and the variability in the relationship between nightlights and emissions, is the custom tuned CH4 inventory used in this work functionally more globally scalable than existing emissions inventories?

Additionally, this paper should include a data availability section as per the ACP data policy: https://www.atmospheric-chemistry-and-physics.net/about/data_policy.html

Specific Comments

Figure 1 & 2: The boundaries drawn on the map in blue and black should be described in the figure captions.

Figure 5: The grey and blue lines should be described in the caption.

P4 L5: Please provide more detail or a citation regarding CO emissions as 1% of CO2.

P5 L18: Is the assumption that the flux from vegetation is balanced based on previous literature?

P13 L1 & P18 L27: Why a factor of 64?

P14 L3: Where is the 20% uncertainty from model winds discussed? If it isn't until the appendix, consider referencing that here.

P25 L24: How was this tuning with OCO-2 observations done?

Technical Corrections

P31 L36: Reference formatted incorrectly

---

## Referee Comment (RC2) · Anonymous Referee #2 · 28 Jun 2018

This paper uses TCCON measurements at Caltech and Lancaster, together with a simple Lagrangian model, to estimate LA fluxes of CO2, methane, and CO. It also applies OCO-2 CO2 data with the same method to estimate CO2 fluxes.

I found the paper to be very informative and thorough, and overall correct as far as I can judge. It does get buried in detail and side alleys and repetitions that make it difficult to read. The authors might consider cutting back unnecessary parts. Also, the advertised premise of the paper is to demonstrate a simple remote sensing method that can be used for estimating urban fluxes worldwide, which a reader might expect to mean using satellite observations, but in fact much of the analysis rests on the TCCON

sites (all of it for methane and CO), and LA is of course an unusually large city which makes the application easier. TCCON is of course "remote sensing", but the title and conclusions may be a little misleading.

The above general comments are simply for the authors to consider at their discretion.

I have a few other minor specific comments below.

1. Introduction: not obvious why one needs top-down estimate of urban fluxes, particularly for CO2 where bottom-up estimates (it seems to me) are likely more reliable. It would be good to give some justification of the need for top-down approaches.

2. Introduction: not clear what the "100+ cities" refers to.

3. Section 2.2: I presume that seasonality of the CO2 flux is neglected since I saw no mention of it. It would be worth making the point that the biospheric term is small in LA, because I wondered about it. Is there also no seasonal pattern in fuel usage?

4. Section 2.4: if the linear inverse model is the way to go why even mention the other two models? Why detail them in the Appendix?

5. Section 4.1: I didn't understand the sentence on "Blooming effects"

---

## Author Comment (AC1) · 12 Oct 2018

**Author responses to Southern California Megacity $CO_2$, $CH_4$, and CO flux estimates using remote sensing and a Lagrangian model**

(https://doi.org/10.5194/acp-2018-517)

Jacob K. Hedelius, Junjie Liu, Tomohiro Oda, Shamil Maksyutov, Coleen M. Roehl, Laura T. Iraci, James R. Podolske, Patrick W. Hillyard, Jianming Liang, Kevin R. Gurney, Debra Wunch, and Paul O. Wennberg

We thank the referees for reviewing this manuscript. Their comments and our responses are below.

Referee comments are in blue with a gray vertical line on the left side.

Our responses are in black.

 Changes to the manuscript are shown with tracked changes in red.

While preparing responses to the referees we also made a necessary improvement to the inversion which altered the flux results. Our revised manuscript reflects these changes.

**R1C1 -** My main concern is regarding the a priori flux estimates used in this work, particularly given the acknowledged existing higher accuracy inventories. I understand that the methodology was designed to be applicable globally, but it is not clear how much the quality of the inversion suffers from this goal.

How different are the CO2 fluxes using the modified ODIAC as compared to using Hestia-LA? Lauvaux et al (2016) used a different Hestia data product and tower measurements in a substantially smaller city; it isn't obvious that the comparison holds over SoCAB with remote sensing data.

We made a sensitivity test using the latest version of Hestia (V2.5). The forward model was more accurate using Hestia V2.5, and the overall flux inversion differed by less than 10%, in agreement with previous studies.

Added to Sect 2.2

As a sensitivity test we also derive a flux based on Hestia-LA 2.5 over the region it is available and Vulcan 3.0 is used for the rest of the area in the U.S. These were gridded to the same scale as the ODIAC.

Added to Sect 4.3

We note that the correlation between the forward model data and TCCON is slightly higher with Hestia than ODIAC, and there are fewer outliers that differ by a factor of 10x or more. However, the flux estimate of $110 \pm 28$ is similar to the posterior flux estimate using ODIAC.

**R1C2 -** Given a lack of information on landfills and the variability in the relationship between nightlights and emissions, is the custom tuned CH4 inventory used in this work functionally more globally scalable than existing emissions inventories?

We have clarified that the CH$_4$ inventory we made in this study is not scalable beyond a national level.

A detailed CH4 inventory is also available for the SoCAB, which we do not use because it would be difficult to scale (Carranza et al., 2018). For the U.S. the Harvard-EPA inventory is already available at 0.1°×0.1° (Maasakkers et al., 2016), and globally the EDGAR inventory is available at 0.1°×0.1° (EC-JRC/PBL, 2009).We make our own 30 arcsec × 30 arcsec methane prior using landfills, nightlights, expected total emissions, and the Harvard-Environmental Protection Agency (EPA) United States (U.S.) inventory (Maasakkers et al., 2016) shown in Fig. 1.  Due to a lack of information outside the U.S. on point sources, such as landfills, our methane prior is also not scalable beyond a national level. For our methane prior we first distribute emissions from landfills as point sources (available 2010–2015, https://ghgdata.epa.gov/ghgp/main.do) and use 2015 emissions for 2016.

**R1C3 -** Additionally, this paper should include a data availability section as per the ACP data policy: https://www.atmospheric-chemistry-and-physics.net/about/data_policy.html

All data used in this study are publically available. No new data were generated. The data availability section is located between the Conclusions section and Appendix (page 18 of the original manuscript).

**R1C4 -** Figure 1 & 2: The boundaries drawn on the map in blue and black should be described in the figure captions.

Thanks. We've added the following to both figure captions:

The black lines are coastlines and the geopolitical boundaries of the SoCAB. Blue lines are county borders.

**R1C5 -** Figure 5: The grey and blue lines should be described in the caption.

We've added the following to the figure caption:

For every 5th sounding the set of backtrajectories is shown in gray. Backtrajectories originating from the AFRC site are shown in blue. Coastlines and the geopolitical boundaries of the SoCAB are shown in black. County borders are shown in blue.

**R1C6 -** P4 L5: Please provide more detail or a citation regarding CO emissions as 1% of CO2.
We added a citation for Wunch et al., 2009.

This same prior is used for CO, but total emissions are 1 % of $CO_2$ emissions on a molar basis (0.6 % of mass) based on the results of Wunch et al. (2009).

**R1C7 -** P5 L18: Is the assumption that the flux from vegetation is balanced based on previous literature?
Estimating the net vegetation flux from the whole basin has been elusive. Due to lack of data some studies are for less than a year, or focus on a few receptor sites. There seems to be a discrepancy for the SoCAB as to whether the biosphere is a net source (Newman et al., 2016) or a net sink (Park et al., 2018). The two studies may not be completely comparable due to different time frames and techniques, but shows that reasonably determining $CO_2$ fluxes from biospheric sources remains a challenge (Feng et al. 2016).

We assume the flux from vegetation is balanced (i.e., no net change in plant biomass or soil carbon) within the basin. This choice is because of uncertainty as to whether there is a net uptake of $CO_2$ by the biosphere in the SoCAB (Park et al., 2018) or if the excess $CO_2$ in the atmosphere from the biosphere (Newman et al., 2016) is due to more respiration than photosynthetic uptake. We estimate the uncertainty due to the biosphere is less than ±10%.

**R1C8 -** P13 L1 & P18 L27: Why a factor of 64?
The factor of 64 for filtering was a somewhat arbitrary choice, but was originally chosen to exclude few observations. Upon reconsidering, we decided on a factor of 10 in this revision due to the possibility of a few outliers strongly affecting results. The sensitivity test in Sect. 4.3 includes changing this factor.

(Sect. 4.3)
We  scale from the starting factor of 10× (Appendix A1).

(Appendix A1)
We also exclude data that differ from the model by a factor of 10 or more. This factor of 10 is somewhat arbitrary and an argument could be made against using this criterion as a filter. However, a few large outliers can significantly affect inversion results (Appendix D2) so we opt to remove suspect values. A sensitivity test including different filter cutoffs for TCCON $X_{CO2}$ is described in Sect. 4.3.  After filtering 2,361 paired OCO-2 - AFRC observations
remain.
For TCCON observations we use the public data, which already has some static within-range
filters applied. We also exclude data that differ from the model by a factor of 10 or greater,
leaving 4,872 observations.
**R1C9 -** P14 L3: Where is the 20% uncertainty from model winds discussed? If it isn't
until the appendix, consider referencing that here.
Yes, the 20% is from the appendix.
… and 20% from model winds (Appendix B).
**R1C10 -** P25 L24: How was this tuning with OCO-2 observations done?
The same way as for the three gases retrieved by TCCON. We've clarified this in the
latest revision.
For simplicity, Sa is chosen as a single scalar value for the linear model (Eq. E3). We
tune two parameters, namely Sa ,
and the threshold for determining linear independence in the
QR decomposition. This is  a trade-off
between maximizing the degrees of freedom and r, avoiding unstable conditions, and minimizing
$\chi 2$. We scan over a variety of Sa and threshold values. We use interannual variability, and
dependence on the prior as noted by a sensitivity test (Fig. 7) to judge the quality. Generally as
we increase the threshold fewer elements are allowed in the state vector, the dependence on the
prior decreases, and the interannual range increases. As Sa increases, so does the interannual
range, and the dependence on the prior decreases. We select values which keep the interannual
variability under about 25%, and minimize dependence on the prior. We repeat this procedure for
the three gases retrieved by TCCON, and for OCO-2 observations. Sa is tuned to 0.01 for $CO_2$,
0.007 for $CH_4$, 0.0007 for CO, and 0.04 for $CO_2$ using OCO-2 observations.
For the 40 factor inversion
Sa is a matrix and diagonal values are
the same as the linear inversion. Off-diagonal values between adjacent elements (e.g., years,
months) are one-third of those along the diagonal , which is a somewhat
arbitrary choice based on our a priori guess of how strongly adjacent elements are related.

**R1C11 -** P31 L36: Reference formatted incorrectly
Fixed, thanks.

**R2C1 -** I found the paper to be very informative and thorough, and overall correct as far as I can judge. It does get buried in detail and side alleys and repetitions that make it difficult to read. The authors might consider cutting back unnecessary parts.
We have reread over the paper with fresh eyes and have tried to better group similar topics and eliminate repetitions. Parts not essential to the central story, but that are required for reproducibility are left in the Appendices.

**R2C2 -** Also, the advertised premise of the paper is to demonstrate a simple remote sensing method that can be used for estimating urban fluxes worldwide, which a reader might expect to mean using satellite observations, but in fact much of the analysis rests on the TCCON sites (all of it for methane and CO), and LA is of course an unusually large city which makes the application easier. TCCON is of course "remote sensing", but the title and conclusions may be a little misleading.
We modified the advertised premise throughout to lessons learned that will be important for future studies estimating urban fluxes worldwide using satellite observations.
The choice of "remote sensing" in the title was to encompass both satellite and TCCON. However, to try to reduce unintentionally misleading readers we have modified the title to:

Southern California Megacity $CO_2$, $CH_4$, and CO flux estimates using ground and space-based remote sensing and a Lagrangian model

**R2C3 -** Introduction: not obvious why one needs top-down estimate of urban fluxes, particularly for CO2 where bottom-up estimates (it seems to me) are likely more reliable. It would be good to give some justification of the need for top-down approaches.
Likely the greatest confidence in emission estimates will be achieved when both bottom-up and top-down approaches agree. Inventories for $CO_2$ are probably much better than say $CH_4$, but there still can be large discrepancies between different inventories (though these might not all be considered "bottom-up"). We've added the following paragraph to the introduction:

Bottom-up inventories (e.g., of $CO_2$) can be derived by accounting for various emission activities such as transportation, electricity generation, industry, and heating. Bottom-up inventories have some inherent uncertainty due to imperfect emission models which are largely based on extrapolation of controlled studies and rely on assumptions of fuel consumption, and from disagreements in downscaling methods (Duren and Miller, 2012; Sargent et al., 2018). Uncertainties in how emissions are calculated and in the underlying activity data used to construct inventories makes them susceptible to systematic biases by nature (Oda et al, 2017). On the national level, 2σ uncertainties range from 4.0-17.5% for the 10 largest emitters (Oda et al, 2018). Uncertainties on the grid cell level are unique to the disaggregation method, but may be in the range of 4—190% (2σ) (Andres et al., 2016). Top-down (TD) emission estimates methods rely on measurements of gases along with models of atmospheric transport, which have their own inherent uncertainties. Measures of emissions, and emission changes are generally more reliable when TD and BU methods are in agreement (Duren and Miller, 2012).

**R2C4 -** Introduction: not clear what the "100+ cities" refers to.
Here "100+ cities" was our way of being quantitative, but it seems to disrupt the flow.

but are too sparse to track emissions from more than a few cities are difficult to scale-up to more than a few dozen areas for long-term observations

**R2C5 -** Section 2.2: I presume that seasonality of the CO2 flux is neglected since I saw no mention of it. It would be worth making the point that the biospheric term is small in LA, because I wondered about it. Is there also no seasonal pattern in fuel usage?
The seasonality in the $CO_2$ prior flux is driven by seasonality in ODAIC. We've added the following:

ODIAC has a monthly variation and compared to the annual average, seasonal flux rates are 1.06 (DJF), 0.97 (MAM), 1.00 (JJA), and 0.97 (SON).

See our response to **R1C7** on the biospheric flux term in LA.

**R2C6 -** Section 2.4: if the linear inverse model is the way to go why even mention the other two models? Why detail them in the Appendix?
Mostly because we were curious as to how they would compare in the end. We think this may be of interest to others in the community so we opted to leave it in.

**R2C7 -** Section 4.1: I didn't understand the sentence on "Blooming effects"

We have revised this part and added a reference.

ODIAC could be low from incorrectly distributing too much of the emissions to rural areas due to . Bblooming effects (Small et al., 2005). (exaggerating the extent of cites Blooming effects refer to the tendency for nightlights to exaggerate settlement areas compared with actual extent due to coarse gridded spatial resolution and indirect or non-electrical lights) in the underlying nightlight data fields in ODAIC could contribute to an incorrect distribution..